# Stabilizing the Q-Gradient Field for Policy Smoothness in Actor–Critic Methods

Jeong Woon Lee [* 1]   Kyoleen Kwak [* 1]   Daeho Kim [* 1]   Hyoseok Hwang [† 1]

## Abstract

Policies learned via continuous actor-critic methods often exhibit erratic, high-frequency oscillations, making them unsuitable for physical deployment. Current approaches attempt to enforce smoothness by directly regularizing the policy's output. We argue that this approach treats the symptom rather than the cause. In this work, we theoretically establish that policy non-smoothness is fundamentally governed by the differential geometry of the critic. By applying implicit differentiation to the actor-critic objective, we prove that the sensitivity of the optimal policy is bounded by the ratio of the Q-function's mixed-partial derivative (noise sensitivity) to its action-space curvature (signal distinctness). To empirically validate this theoretical insight, we introduce *Policy-Aware Value-field Equalization* (PAVE), a critic-centric regularization framework that treats the critic as a scalar field and *stabilizes* its induced action-gradient field. PAVE rectifies the learning signal by minimizing the Q-gradient volatility while preserving local curvature. Experimental results demonstrate that PAVE achieves smoothness comparable to policy-side smoothness regularization methods, while maintaining competitive task performance, without modifying the actor.

## 1. Introduction

Deep reinforcement learning (RL) has become a standard approach for continuous control, with strong results across locomotion and manipulation benchmarks (Hwangbo et al., 2019; Levine et al., 2016) using actor–critic methods such as Soft Actor-Critic (SAC) (Haarnoja et al., 2018) and Twin Delayed Deep Deterministic Policy Gradient (TD3) (Fujimoto et al., 2018). However, a persistent gap between

simulation success and real deployment remains action instability: trained policies often exhibit high-frequency oscillations and "jerky" control signals (Chen et al., 2021; Mysore et al., 2021). In physical systems, such oscillations translate into higher energy consumption, accelerated actuator wear, vibration-induced fatigue, and potential safety issues; (Inman, 2013; de la Presilla et al., 2023) they also degrade human-perceived quality even when task return is high (Chen et al., 2021).

To mitigate this, the dominant paradigm has been to explicitly constrain the policy network. This body of work includes adding regularization terms to the actor's loss function to enforce spatial or temporal consistency (Mysore et al., 2021; Kobayashi, 2022; Lee et al., 2024), and more recently, aligning actions with trajectory-based predictions to capture system dynamics (Kwak & Hwang, 2026). Alternatively, architectural approaches modify the policy network structure to enforce Lipschitz continuity (Takase et al., 2022; Song et al., 2023). Theoretically, these methods share a common goal of reducing the policy network's Lipschitz constant (Song et al., 2023).

While these approaches mitigate action oscillations, we contend that they address the *symptom* rather than the *cause*. Even under strong regularization, if the underlying Q-function remains noisy, the resulting policy gradients continue to drive the actor in erratic directions. This introduces a conflict during optimization: the actor faces competing objectives—minimizing the smoothing penalty versus ascending a volatile learning signal. Consequently, the policy is compelled to diverge from the return-maximizing manifold to satisfy smoothness constraints, compromising task performance.

We propose a fundamental shift in perspective: from forcing the actor to be smooth, to inducing smoothness via a stable learning signal. We argue that the root cause of policy non-smoothness lies deeper, within the geometric structure of the critic's value function that guides the actor. This dynamic is central to continuous actor-critic frameworks, which have become the standard for continuous control (Lillicrap et al., 2016; Fujimoto et al., 2018; Haarnoja et al., 2018). In this paradigm, the policy is updated by ascending the gradient of the Q-function with respect to the action, $\nabla_a Q(s, a)$. This Q-gradient field serves as the learning signal, defining the

---

[*]Equal contribution   [1]College of Software, Kyung Hee University.   Correspondence to: Hyoseok Hwang <hyoseok@khu.ac.kr>.

*Proceedings of the 43ʳᵈ International Conference on Machine Learning*, Seoul, South Korea. PMLR 306, 2026. Copyright 2026 by the author(s).

direction of steepest ascent for policy improvement. If this field is geometrically unstable—where the gradient direction flips drastically with infinitesimal state perturbations—the actor is fundamentally misled. Since the policy is optimized to follow these gradients, diverging optimization directions at neighboring states compel the actor to produce disparate actions, directly manifesting as high-frequency oscillations.

To formalize this idea, we analyze the optimal policy induced by the critic via implicit differentiation. This theoretical examination reveals that the sensitivity of the optimal action is strictly governed by the interplay of two geometric properties: (i) the mixed state–action Hessian $\nabla^2_{sa}Q$ and (ii) the conditioning of the action Hessian $\nabla^2_{aa}Q$. The mixed Hessian $\nabla^2_{sa}Q$ quantifies the volatility of the learning signal, representing how rapidly the direction of steepest ascent rotates with small state variations. Meanwhile, the action Hessian $\nabla^2_{aa}Q$ dictates the curvature of the peak itself; a flat landscape creates an ill-conditioned maximization problem where the optimal action becomes hypersensitive to negligible shifts.

This observation suggests a different strategy: rather than constraining the actor to compensate for an irregular learning signal, we propose to directly regularize the geometry of the Q-function itself. Intuitively, if the value field is smooth and well-conditioned over the state–action manifold, the induced action-gradient field becomes stable by construction. Crucially, this approach differs from naively smoothing the value function, which might obscure necessary control information. By minimizing noise sensitivity ($\nabla^2_{sa}Q$) while explicitly preserving signal curvature ($\nabla^2_{aa}Q$), we ensure that smooth policies emerge naturally without sacrificing the distinctness of the optimal action.

To empirically validate this theoretical insight, we introduce *Policy-Aware Value-field Equalization* (PAVE), a critic-centric regularization framework that treats the critic as a scalar field over the state–action manifold and *stabilizes* its induced action-gradient field. PAVE combines (i) *Mixed-Partial Regularization* to bound state-induced changes of the action-gradient (ii) *Vector Field Consistency* based on Fisher divergence to align Q-gradient fields across temporal transitions, and (iii) *Curvature Preservation* to maintain well-defined local maxima. Our experimental results show that, by modifying the critic's geometric structure alone, PAVE achieves action smoothness comparable to policy-side regularization methods, without any modification to the actor. Our contributions are as follows:

- We identify the critic's geometry as the primary source of action instability. By applying implicit differentiation, we rigorously prove that policy sensitivity is fundamentally governed by the mixed-partial derivatives and action curvature of the Q-function.

- We propose PAVE, a critic-centric regularization framework designed to validate our theory by stabilizing the Q-gradient field rather than constraining the actor.

- We provide empirical evidence that *paving* the value landscape yields enhanced control smoothness while maintaining competitive task scores, thereby validating our hypothesis that critic geometry is the fundamental driver of action stability.

## 2. Related Work

### 2.1. Policy-side smoothness regularization

A prevalent strategy for mitigating action oscillations augments the actor's objective with regularization terms that limit the policy network's Lipschitz constant. A seminal work in this domain, Conditioning for Action Policy Smoothness (CAPS) (Mysore et al., 2021), penalizes differences between temporally consecutive actions and spatially adjacent states sampled from a fixed Gaussian neighborhood. To improve adaptability, Local Lipschitz Continuous Constraint (L2C2) (Kobayashi, 2022) dynamically adjusts the spatial neighborhood radius based on the state. Gradient-based CAPS (Grad-CAPS) (Lee et al., 2024) regularizes action gradients to suppress high-frequency zigzagging. Most recently, ASAP (Kwak & Hwang, 2026) refines spatial regularization by utilizing transition-induced distributions and aligning actions with predictions from preceding states, supplemented by second-order temporal penalties. Complementary to loss-based penalties, architectural methods modify the neural network structure to intrinsically satisfy Lipschitz continuity. Spectral Normalization (Gogianu et al., 2021; Bjorck et al., 2021) has been adapted to RL to stabilize training by bounding the policy's Lipschitz constant. More specialized architectures, LipsNet (Song et al., 2023) and its extension LipsNet++ (Song et al., 2025), constrain the policy's Jacobian via Multi-dimensional Gradient Normalization (MGN) and Fourier-based filtering, respectively. Action chunking (Zhao et al., 2023) achieves temporal consistency by predicting action sequences at the output level; in contrast, PAVE operates at the critic level, stabilizing the learning signal that drives actor updates.

### 2.2. Critic geometry and gradient regularization

Stabilizing learning signals via gradient/Jacobian regularization has been explored broadly (Sokolić et al., 2017; Roth et al., 2017). In RL, these methods are typically applied to improve robustness or training stability, but they do not explicitly connect critic differential geometry to policy smoothness (Gogianu et al., 2021). Our work focuses on the specific geometric terms that govern the state-sensitivity of the critic-induced greedy policy (mixed partials and action curvature), and proposes targeted regularization that

minimizes volatility while explicitly preserving useful local discrimination. In our approach, this can be interpreted as stabilizing and equalizing the induced *Q-gradient field* that the actor follows during learning.

## 2.3. Score matching and Fisher divergence

Score matching aligns gradients of log-densities without partition functions (Hyvärinen, 2005). Modern score-based generative modeling learns smooth score fields with related objectives (Song & Ermon, 2019). We adapt this perspective to RL by treating $\nabla_a Q$ as a vector field whose temporal consistency can be regularized, yielding more stable actor update directions along trajectories.

## 3. Preliminary

We formulate the continuous control problem as a Markov Decision Process (MDP) defined by the tuple $(\mathcal{S}, \mathcal{A}, P, r, \gamma)$, where $\mathcal{S}$ and $\mathcal{A}$ denote the continuous state and action spaces, $P : \mathcal{S} \times \mathcal{A} \to \mathcal{P}(\mathcal{S})$ is the transition dynamics, $r : \mathcal{S} \times \mathcal{A} \to \mathbb{R}$ is the reward function, and $\gamma \in [0, 1)$ is the discount factor. The objective is to learn a policy $\pi : \mathcal{S} \to \mathcal{P}(\mathcal{A})$ that maximizes the expected return $\mathbb{E}[\sum_{t=0}^{\infty} \gamma^t r(s_t, a_t)]$.

Modern actor–critic methods learn a critic $Q_\theta(s, a)$ to approximate the action-value function and update a parameterized policy $\pi_\phi$ by ascending the Q-function's gradient. For deterministic actors, such as TD3 (Fujimoto et al., 2018), the update follows the Deterministic Policy Gradient (DPG) (Silver et al., 2014):

$$\nabla_\phi J(\phi) = \mathbb{E}_{s \sim \mathcal{D}} \left[ \nabla_a Q_\theta(s, a)|_{a=\pi_\phi(s)} \nabla_\phi \pi_\phi(s) \right], \quad (1)$$

where $\mathcal{D}$ represents the replay buffer. Similarly, for stochastic actors exemplified by SAC (Haarnoja et al., 2018), although the objective includes an entropy term, the policy update is fundamentally driven by the same gradient flow $\nabla_a Q_\theta(s, a)$ via the reparameterization trick.

In both paradigms, the actor attempts to approximate the implicit greedy policy $a^*(s) = \arg\max_a Q_\theta(s, a)$. Consequently, the geometric properties of the critic directly dictate the stability of the actor. We treat the gradient $\nabla_a Q_\theta(s, \cdot)$ as a vector field over the state–action manifold; stabilizing the geometry of Q-gradient field is the core focus of PAVE.

## 4. Theoretical Analysis: The Geometry of Policy Sensitivity

To remediate the fundamental origin of action instability in actor-critic methods, it is imperative to analyze how the differential geometry of the value function dictates the behavior of the induced policy. In this section, we rigorously derive the sensitivity of the greedy policy

$a^*(s) = \arg\max_a Q_\theta(s, a)$ with respect to state perturbations. By integrating the Implicit Function Theorem (IFT) into our analysis, we explicitly demonstrate that the smoothness of the policy is governed by specific Hessian terms of the critic.

### 4.1. Implicit Policy Definition and Sensitivity Derivation

In the actor-critic paradigm, the actor $\pi_\phi(s)$ is optimized to approximate the maximizer of the Q-function. Even in continuous control settings where the actor is a neural network, its gradient update direction is fundamentally driven by $\nabla_a Q_\theta(s, \pi_\phi(s))$. Consequently, the geometric regularity of the explicit maximizer $a^*(s) = \arg\max_a Q_\theta(s, a)$ imposes a fundamental limit on the policy: the parameterized actor $\pi_\phi$ cannot exceed the smoothness of $a^*$ without deviating from the optimal action and incurring a performance penalty. We formally characterize the local sensitivity of the optimal action via the following lemma.

**Lemma 4.1** (Implicit Policy Jacobian). *Let $Q_\theta : \mathcal{S} \times \mathcal{A} \to \mathbb{R}$ be a twice continuously differentiable function. Assume that for a given state $s$, $a^*(s)$ corresponds to a strict local maximum and is an interior point of $\mathcal{A}$, such that the action Hessian $\nabla_{aa}^2 Q_\theta(s, a^*(s))$ is negative definite. Then, the sensitivity of the optimal action with respect to the state, denoted by the policy Jacobian $J_\pi(s) = \nabla_s a^*(s)$, is analytically given by:*

$$J_\pi(s) = - \left[ \nabla_{aa}^2 Q_\theta(s, a^*(s)) \right]^{-1} \nabla_{sa}^2 Q_\theta(s, a^*(s)). \quad (2)$$

*Proof.* Since $a^*(s)$ is an interior extremum, it must satisfy the first-order optimality condition $\nabla_a Q_\theta(s, a^*(s)) = 0$. We consider this gradient as a vector-valued mapping $G(s, a(s)) = 0$. Applying the total derivative with respect to $s$ via the chain rule yields:

$$\begin{aligned}
\frac{d}{ds} G(s, a^*(s)) &= \nabla_{sa}^2 Q_\theta(s, a^*(s)) \\
&+ \nabla_{aa}^2 Q_\theta(s, a^*(s)) \nabla_s a^*(s) = 0.
\end{aligned} \quad (3)$$

Given the strict concavity assumption, the Hessian $\nabla_{aa}^2 Q_\theta(s, a^*(s))$ is invertible. Rearranging the terms, we obtain:

$$\nabla_{aa}^2 Q_\theta(s, a^*(s)) \nabla_s a^*(s) = -\nabla_{sa}^2 Q_\theta(s, a^*(s)). \quad (4)$$

Pre-multiplying by the inverse Hessian $\left[ \nabla_{aa}^2 Q_\theta(s, a^*(s)) \right]^{-1}$ yields the expression in Equation (2). $\square$

*Remark* 4.2. Lemma 4.1 requires $a^*(s)$ to be an interior strict local maximum. When optimal actions lie on the boundary of $\mathcal{A}$ or are not unique, the implicit function theorem does not apply and $J_\pi(s)$ may not be well-defined.

In practice, continuous-control environments with tanh-bounded actions rarely saturate the boundary, and our empirical results (Sec. 6) confirm that the regularization motivated by this analysis is effective even when the assumption is not universally satisfied.

Equation (2) offers a crucial geometric intuition: policy sensitivity is the product of the *inverse curvature* and the *mixed-partial coupling*. Intuitively, $\nabla^2_{sa} Q_\theta$ acts as a forcing term that dictates how the ascent direction shifts, while $[\nabla^2_{aa} Q_\theta]^{-1}$ acts as an amplification factor determined by the landscape's flatness. If the Q-surface is flat (low curvature), the inverse Hessian explodes, rendering the policy hypersensitive to even negligible gradient rotations. This factorization provides the rigorous mathematical basis for our proposed regularization strategy.

### 4.2. Spectral Bounds on Lipschitz Continuity

We now establish a concrete bound on the Lipschitz constant of the policy by analyzing the spectral norms of the constituent matrices in Lemma 4.1.

**Proposition 4.3** (Lipschitz Continuity Bound). *Let $\| \cdot \|_2$ denote the spectral norm. Suppose the mixed partial derivative is bounded by $\|\nabla^2_{sa} Q_\theta(s, a^*(s))\|_2 \leq M$ and the action Hessian satisfies the strict concavity condition $\lambda_{\max}(\nabla^2_{aa} Q_\theta(s, a^*(s))) \leq -\mu < 0$ (where $\lambda_{\max}(\cdot)$ denotes the maximum eigenvalue). Then, the induced greedy policy is Lipschitz continuous with a constant $L$ satisfying:*

$$\|a^*(s) - a^*(s')\|_2 \leq L\|s - s'\|_2, \quad \text{where } L \leq \frac{M}{\mu}. \quad (5)$$

*Proof.* We analyze the spectral norm of the policy Jacobian derived in Theorem 4.1. Leveraging the sub-multiplicative property of the spectral norm, we have:

$$\|J_\pi(s)\|_2 \leq \left\| \left[\nabla^2_{aa} Q_\theta(s, a^*(s))\right]^{-1} \right\|_2 \cdot \left\| \nabla^2_{sa} Q_\theta(s, a^*(s)) \right\|_2. \quad (6)$$

**1. Curvature Bound:** The assumption $\lambda_{\max}(\nabla^2_{aa} Q_\theta(s, a^*(s))) \leq -\mu$ implies that the spectrum of the Hessian lies entirely in $(-\infty, -\mu]$. Consequently, the eigenvalues of the inverse Hessian are bounded in magnitude by $1/\mu$. Thus:

$$\left\| \left[\nabla^2_{aa} Q_\theta\right]^{-1} \right\|_2 = \frac{1}{|\lambda_{\max}(\nabla^2_{aa} Q_\theta(s, a^*(s)))|} \leq \frac{1}{\mu}. \quad (7)$$

**2. Mixed-Partial Bound:** By hypothesis, the sensitivity of the gradient field to state variations is bounded by $\|\nabla^2_{sa} Q_\theta\|_2 \leq M$.

Substituting these bounds, we define the Lipschitz constant $L$ as the supremum of the policy Jacobian norm:

$$L \triangleq \sup_s \|\nabla_s a^*(s)\|_2 \leq \frac{M}{\mu}. \quad (8)$$

Finally, applying the Mean Value Inequality for vector-valued differentiable functions, for any pair of states $s, s'$:

$$\begin{aligned}
&\|a^*(s) - a^*(s')\|_2 \\
&\leq \left( \sup_z \|\nabla_s a^*(z)\|_2 \right) \|s - s'\|_2 \\
&= L\|s - s'\|_2 \leq \frac{M}{\mu}\|s - s'\|_2.
\end{aligned} \quad (9)$$

This confirms that the policy is Lipschitz continuous with constant $L \leq M/\mu$. $\square$

*Remark* 4.4. The condition $\lambda_{\max}(\nabla^2_{aa} Q_\theta(s, a^*(s))) \leq -\mu < 0$ may not hold everywhere for neural network critics. In our experiments, we find that unconstrained critics satisfy this condition in only 14–47% of visited states (see Sec. 6). The bound $L \leq M/\mu$ therefore serves as regularization motivation rather than a universal guarantee. PAVE's $\mathcal{L}_{\text{Curv}}$ improves satisfaction to 64–100%.

**Theoretical Implication.** The derived bound $L \leq M/\mu$ highlights a critical insight: simply minimizing the gradient variance (reducing $M$) is insufficient if the curvature $\mu$ also vanishes. Conventional regularization methods often inadvertently flatten the Q-landscape, driving $\mu \to 0$, which can theoretically explode the sensitivity term $\|(\nabla^2_{aa} Q_\theta(\cdot))^{-1}\|$. PAVE is explicitly formulated to minimize $M$ while ensuring $\mu$ remains strictly bounded away from zero.

## 5. PAVE: Policy-Aware Value-field Equalization

We introduce PAVE, a critic-centric framework designed to explicitly enforce the geometric stability conditions derived in Section 4. Instead of constraining the actor, PAVE directly regularizes the value field to minimize the Lipschitz bound $L \leq M/\mu$ and enforce trajectory consistency. This is achieved via three synergistic objectives: (1) suppressing noise sensitivity ($M$) via *Mixed-Partial Regularization*, (2) aligning temporal vector fields via *Vector Field Consistency*, and (3) preserving curvature ($\mu$) via *Curvature Preservation* to prevent geometric collapse. The following losses are finite-difference proxies that incentivize the desired geometric properties rather than guaranteeing them (see the Remarks in Sec. 4).

### 5.1. Mixed-Partial Regularization (MPR)

To minimize the numerator $M$ in Equation (5), it is necessary to suppress the magnitude of the mixed partial derivatives $\nabla^2_{sa} Q_\theta$. However, explicit computation of the Hessian imposes a computational complexity of $\mathcal{O}(d^2)$, which is prohibitive for online RL.

We circumvent this by employing a finite-difference approximation grounded in Taylor expansion. Consider the pertur-

bation of the action-gradient $\nabla_a Q_\theta$ induced by a small state variation $\epsilon$:

$$\nabla_a Q_\theta(s+\epsilon, a) = \nabla_a Q_\theta(s, a) + \nabla^2_{sa} Q_\theta(s, a)\epsilon + \mathcal{O}(\|\epsilon\|^2). \tag{10}$$

Neglecting higher-order terms, the difference $\|\nabla_a Q_\theta(s+\epsilon, a) - \nabla_a Q_\theta(s, a)\|$ serves as an efficient proxy for the Hessian-vector product $\|\nabla^2_{sa} Q_\theta(s, a)\epsilon\|$. Accordingly, we define the Mixed-Partial Regularization (MPR) objective as:

$$\mathcal{L}_{\text{MPR}}(\theta) = \mathbb{E}_{\substack{(s,a)\sim\mathcal{D}\\\epsilon\sim\mathcal{N}(0,\sigma^2 I)}} \Big[ \big\|\nabla_a Q_\theta(s+\epsilon, a) \\ - \nabla_a Q_\theta(s, a)\big\|_2^2 \Big]. \tag{11}$$

Minimizing this objective enforces local invariance of the vector field $\nabla_a Q_\theta$ with respect to state perturbations, effectively compressing the spectral norm $M$ and widening the region of attraction for the optimal action.

**Formal Justification.** We further provide a mathematical justification for why minimizing Equation (11) encourages a smaller $M$. By substituting the linear approximation into the objective and taking the expectation over the isotropic Gaussian noise $\epsilon$, we obtain:

$$\begin{aligned} \mathcal{L}_{\text{MPR}} &\approx \mathbb{E}_\epsilon \Big[ \big\|\nabla^2_{sa} Q_\theta(s, a)\epsilon\big\|_2^2 \Big] \\ &= \mathbb{E}_\epsilon \big[ \epsilon^\top (\nabla^2_{sa} Q_\theta(s, a))^\top \nabla^2_{sa} Q_\theta(s, a)\epsilon \big] \\ &= \sigma^2 \big\|\nabla^2_{sa} Q_\theta(s, a)\big\|_F^2 . \end{aligned} \tag{12}$$

Since the spectral norm is upper-bounded by the Frobenius norm ($\|A\|_2 \leq \|A\|_F$), minimizing $\mathcal{L}_{\text{MPR}}$ encourages a smaller upper bound on $M$. Note that this is a finite-difference proxy that incentivizes the desired geometric property rather than guaranteeing it.

## 5.2. Vector Field Consistency (VFC)

While MPR enforces spatial smoothness via isotropic perturbations, stable robotic control necessitates *temporal* coherence along trajectories. Ideally, the learning signal driving the actor should not exhibit high-frequency fluctuations between consecutive time steps (Kwak & Hwang, 2026).

To formalize this, we draw inspiration from Score Matching (Hyvärinen, 2005). By interpreting the gradient $\nabla_a Q_\theta(s, a)$ as the score function of an implicit Boltzmann policy $p(a|s) \propto \exp(Q_\theta(s, a))$, we frame temporal stability as minimizing the distributional shift of the policy. Specifically, we minimize the Euclidean distance between the score fields induced by consecutive states $s_t$ and $s_{t+1}$, which is analogous to minimizing the Fisher Divergence:

$$\mathcal{L}_{\text{VFC}}(\theta) = \mathbb{E}_{(s_t,a_t,s_{t+1})\sim\mathcal{D}} \Big[ \big\|\nabla_a Q_\theta(s_t, a_t) \\ - \nabla_a Q_\theta(s_{t+1}, a_t)\big\|_2^2 \Big]. \tag{13}$$

---

**Algorithm 1** PAVE: Policy-Aware Value-field Equalization

1: **Input:** Hyperparameters $\lambda_1, \lambda_2, \lambda_3$, noise scale $\sigma$, margin $\delta$
2: **Initialize:** Critic $\theta$, Actor $\phi$, Target networks $\theta', \phi'$, Replay buffer $\mathcal{D}$
3: **for** each environment step **do**
4:      Sample action $a_t \sim \pi_\phi(s_t)$ (with exploration noise)
5:      Execute $a_t$, observe $r_t, s_{t+1}$, store $(s_t, a_t, r_t, s_{t+1})$ in $\mathcal{D}$
6:      **for** each gradient step **do**
7:          Sample batch $\mathcal{B} = \{(s, a, r, s')\} \sim \mathcal{D}$
8:          // 1. Standard TD Learning
9:          Compute $\mathcal{L}_{\text{TD}}(\theta)$ using targets $\theta', \phi'$
10:         // 2. Mixed-Partial Regularization (MPR)
11:         Sample perturbations $\epsilon \sim \mathcal{N}(\mathbf{0}, \sigma^2 I)$
12:         $\mathcal{L}_{\text{MPR}} \leftarrow \frac{1}{|\mathcal{B}|} \sum \|\nabla_a Q_\theta(s+\epsilon, a) - \nabla_a Q_\theta(s, a)\|_2^2$
13:         // 3. Vector Field Consistency (VFC)
14:         $\mathcal{L}_{\text{VFC}} \leftarrow \frac{1}{|\mathcal{B}|} \sum \|\nabla_a Q_\theta(s, a) - \nabla_a Q_\theta(s', a)\|_2^2$
15:         // 4. Curvature Preservation (Curv)
16:         Sample vectors $v \sim$ Rademacher$(|\mathcal{A}|)$
17:         $Tr_{est} \leftarrow v^\top \nabla^2_{aa} Q_\theta(s, a) v$
18:         $\mathcal{L}_{\text{Curv}} \leftarrow \frac{1}{|\mathcal{B}|} \sum \max(0, Tr_{est} + \delta)$
19:         // 5. Joint Update
20:         $\mathcal{L}_{\text{Total}} \leftarrow \mathcal{L}_{\text{TD}} + \lambda_1 \mathcal{L}_{\text{MPR}} + \lambda_2 \mathcal{L}_{\text{VFC}} + \lambda_3 \mathcal{L}_{\text{Curv}}$
21:         $\theta \leftarrow \theta - \eta \nabla_\theta \mathcal{L}_{\text{Total}}$
22:         Update actor $\phi$ via $\nabla_\phi J(\phi)$
23:      **end for**
24: **end for**

---

This objective aligns the direction of policy improvement along the temporal dimension, effectively mitigating the "chattering" effect caused by conflicting gradients at adjacent time steps.

**Formal Justification.** Similar to MPR, VFC also encourages a smaller $M$, but specifically along dynamics-relevant directions. Considering the first-order expansion $s_{t+1} \approx s_t + \Delta s_t$, the objective approximates the directional Hessian norm:

$$\begin{aligned} \|\nabla_a Q_\theta(s_{t+1}, a_t) - \nabla_a Q_\theta(s_t, a_t)\|_2^2 &\approx \\ \|\nabla^2_{sa} Q_\theta(s_t, a_t) \Delta s_t\|_2^2. \end{aligned} \tag{14}$$

While MPR reduces $M$ globally, VFC specifically encourages smaller $M$ along the state transitions imposed by the environment dynamics ($\nabla^2_{sa} Q_\theta(\Delta)s_t$).

## 5.3. Curvature Preservation (Curv)

Minimizing $\mathcal{L}_{\text{MPR}}$ in isolation introduces a pathological risk: the neural network may collapse the Q-function to a trivial flat plane (i.e., $\nabla_a Q_\theta \approx \mathbf{0}, \forall s, a$) to satisfy the smoothness constraint. As established in Proposition 4.3,

*Table 1.* Cumulative return (*re*) and smoothness score (*sm*) on Gymnasium benchmark under TD3 setting (SiLU-unified, 5 seeds). Higher *re* and lower *sm* are better. Bold indicates best, and underlined the second-best, per environment. Standard deviations over 5 seeds shown in parentheses.

| Method | LunarLander | | Pendulum | | Reacher | | Ant | | Hopper | | Walker | |
|---|---|---|---|---|---|---|---|---|---|---|---|---|
| | *re* ↑ | *sm* ↓ | *re* ↑ | *sm* ↓ | *re* ↑ | *sm* ↓ | *re* ↑ | *sm* ↓ | *re* ↑ | *sm* ↓ | *re* ↑ | *sm* ↓ |
| TD3 Base | 227.8 (74.1) | 1.809 (1.311) | -168.7 (77.9) | 1.590 (0.571) | -3.55 (1.32) | 0.053 (0.014) | 4299 (1499) | 2.039 (0.400) | 3056 (888) | 2.715 (0.483) | 4834 (423) | 1.990 (0.208) |
| CAPS | 249.0 (40.3) | 0.702 (0.291) | -175.2 (79.3) | 0.464 (0.165) | -3.58 (1.35) | 0.047 (0.011) | 4624 (1190) | 2.135 (0.256) | **3609** (33) | 1.919 (0.373) | 4913 (1383) | 1.600 (0.306) |
| GRAD | 245.1 (73.4) | 0.669 (0.184) | **-167.6** (75.6) | 0.689 (0.139) | **-3.55** (1.36) | 0.041 (0.012) | **5538** (845) | 1.796 (0.187) | 3317 (648) | 1.642 (0.405) | 4788 (783) | 1.366 (0.194) |
| ASAP | 168.4 (147.0) | 2.055 (1.300) | -170.2 (80.7) | 1.970 (0.768) | -3.56 (1.35) | 0.051 (0.014) | 4765 (1554) | 2.065 (0.413) | 3229 (553) | 1.991 (0.259) | 4740 (799) | 1.616 (0.284) |
| PAVE | **264.5** (24.9) | **0.541** (0.290) | -167.6 (77.5) | **0.351** (0.118) | -4.02 (1.34) | **0.039** (0.013) | 4649 (1443) | **1.768** (0.315) | 3305 (461) | **0.950** (0.285) | **5563** (451) | **1.272** (0.172) |

if $Q_\theta$ becomes flat, the term $\|(\nabla^2_{aa} Q_\theta(s, a))^{-1}\|$ diverges, paradoxically amplifying policy sensitivity.

To preclude this "over-smoothing" collapse, we must enforce the geometric constraint that the curvature $\mu$ remains positive. We propose Curvature Preservation (Curv), which penalizes the trace of the action Hessian if it fails to maintain sufficient concavity:

$$\mathcal{L}_{\text{Curv}} = \mathbb{E}_{\substack{(s,a)\sim\mathcal{D} \\ v\sim p(v)}} \left[ \max\left(0,\ v^\top \nabla^2_{aa} Q_\theta(s, a) v + \delta\right) \right], \tag{15}$$

where $\delta > 0$ defines the minimum required sharpness. To ensure computational efficiency, we employ Hutchinson's trace estimator with random Rademacher vectors $v$. Note that we utilize SiLU activation in the critic to ensure $C^2$ continuity for valid Hessian computation.

**Formal Justification.** This regularizer explicitly targets the denominator $\mu$ of the bound. By penalizing projected curvature values $v^\top \nabla^2_{aa} Q_\theta(s, a) v$ that exceed $-\delta$, this objective incentivizes concavity via Hutchinson's trace estimator, which controls the trace rather than individual eigenvalues. This encourages the maximum eigenvalue to remain negative, helping to keep the inverse Hessian norm bounded and reducing sensitivity.

### 5.4. Total Objective Formulation

The composite objective function for the critic parameter $\theta$ is formulated as a weighted sum of the TD error and the proposed geometric regularizers:

$$\mathcal{L}(\theta) = \mathcal{L}_{\text{TD}} + \lambda_1 \mathcal{L}_{\text{MPR}} + \lambda_2 \mathcal{L}_{\text{VFC}} + \lambda_3 \mathcal{L}_{\text{Curv}}, \tag{16}$$

where $\mathcal{L}_{\text{TD}}$ denotes the standard temporal-difference regression objective. The complete training procedure is summarized in Algorithm 1. Crucially, these geometric regularizers are applied solely as auxiliary losses to the critic. Consequently, the actor update mechanism remains unchanged,

utilizing unmodified standard policy gradients while benefiting significantly from the paved, well-conditioned Q-gradient landscape.

## 6. Empirical Analysis

### 6.1. Experimental Setup

Experiments were conducted using Gymnasium (Towers et al., 2024) and MuJoCo (Todorov et al., 2012) across six environments: *LunarLanderContinuous-v2*, *Pendulum-v1*, *Reacher-v2*, *Ant-v5*, *Hopper-v5*, and *Walker2d-v5*. This suite spans low-dimensional stabilization to high-dimensional articulated locomotion, providing a diverse testbed for evaluating both task performance and policy smoothness. For evaluation metrics, we use Cumulative Return (*re*) to assess task competency and Smoothness Score (*sm*) to quantify action oscillations. The latter is a spectral smoothness metric derived via Fast Fourier Transform (FFT) analysis, following established protocols (Mysore et al., 2021; Christmann et al., 2024). We built the codebase on top of Stable-Baselines3 (Raffin et al., 2021), building upon the implementation provided in (Kwak & Hwang, 2026). We integrated PAVE into two widely-used off-policy algorithms, Twin Delayed Deep Deterministic policy gradient (TD3) (Fujimoto et al., 2018) and Soft Actor-Critic (SAC) (Haarnoja et al., 2018). Regarding network architectures, all methods use SiLU (Elfwing et al., 2018) activation for fair comparison. SiLU is a $C^2$-continuous approximation of ReLU, which is required by $\mathcal{L}_{\text{Curv}}$ for valid Hessian computation via Hutchinson's trace estimator. We trained our method with five independent random seeds.

### 6.2. Analysis of Geometric Regularization

In TD3 (Table 1), our results validated the theoretical premise that stabilizing the Q-gradient field can naturally induce policy smoothness without sacrificing perfor-

*Table 2.* Cumulative return (*re*) and smoothness score (*sm*) on Gymnasium benchmark under SAC setting (SiLU-unified, 5 seeds). Higher *re* and lower *sm* are better. Bold indicates best, and underlined the second-best, per environment. Standard deviations over 5 seeds shown in parentheses.

| Method | LunarLander | | Pendulum | | Reacher | | Ant | | Hopper | | Walker | |
|---|---|---|---|---|---|---|---|---|---|---|---|---|
| | $re\uparrow$ | $sm\downarrow$ | $re\uparrow$ | $sm\downarrow$ | $re\uparrow$ | $sm\downarrow$ | $re\uparrow$ | $sm\downarrow$ | $re\uparrow$ | $sm\downarrow$ | $re\uparrow$ | $sm\downarrow$ |
| SAC Base | 160.3 (135.3) | 0.434 (0.187) | **-163.1** (74.5) | 0.548 (0.173) | -3.73 (1.37) | 0.053 (0.016) | 5276 (1029) | 1.941 (0.289) | 3481 (98) | 0.773 (0.069) | 4907 (283) | 0.748 (0.063) |
| CAPS | **270.9** (19.6) | 0.271 (0.051) | -166.1 (76.9) | 0.338 (0.118) | -3.69 (1.32) | 0.048 (0.014) | 5447 (884) | 1.843 (0.241) | 3442 (52) | 0.660 (0.110) | 4689 (178) | 0.776 (0.071) |
| GRAD | 256.6 (38.2) | 0.252 (0.063) | -163.7 (74.5) | 0.377 (0.111) | -3.74 (1.29) | **0.046** (0.013) | **5729** (604) | 1.632 (0.103) | 3315 (407) | 0.581 (0.072) | 4887 (211) | 0.572 (0.052) |
| ASAP | 268.6 (20.8) | 0.183 (0.033) | -163.5 (74.7) | 0.362 (0.109) | **-3.66** (1.31) | 0.048 (0.015) | 5707 (852) | **1.404** (0.088) | 3077 (715) | **0.404** (0.059) | 4731 (209) | **0.557** (0.069) |
| PAVE | 265.8 (21.4) | **0.142** (0.028) | -165.4 (75.5) | **0.290** (0.130) | -3.71 (1.31) | 0.052 (0.014) | 5706 (774) | 1.604 (0.146) | **3489** (38) | 0.556 (0.139) | **4954** (250) | 0.584 (0.020) |

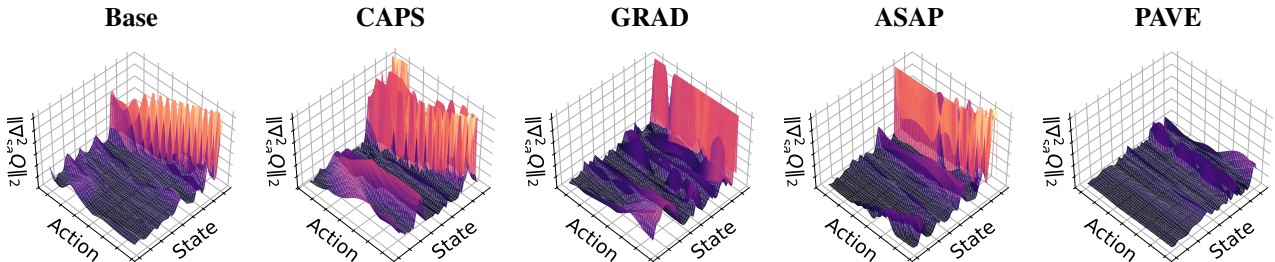

**Base**  **CAPS**  **GRAD**  **ASAP**  **PAVE**

*Figure 1.* 3D visualization of the mixed-partial Hessian spectral norm $\|\nabla^2_{sa}Q\|_2$ in Walker2d (SiLU-unified, autograd exact). All methods share the same color scale (Z-axis clipped at Base 99th percentile). PAVE effectively stabilizes the Q-gradient field compared to baselines.

mance. PAVE achieved the best smoothness across all six environments—with the most dramatic reductions on Hopper ($2.715 \to 0.950$, $2.9\times$) and Pendulum ($1.590 \to 0.351$, $4.5\times$)—while maintaining or improving Base returns in 5/6 environments. On Walker, PAVE simultaneously secured the highest cumulative return ($4834 \to 5563$) and the best smoothness score ($1.990 \to 1.272$). This indicates that for deterministic policies, which are prone to overfitting to sharp gradient irregularities (Matheron et al., 2019), rectifying the critic's geometry is a highly effective strategy.

The SAC setting in Table 2 reveals environment-dependent trade-offs that complement the TD3 findings. PAVE achieved the best smoothness on LunarLander ($0.142$ vs Base $0.434$) and Pendulum ($0.290$ vs Base $0.548$); on Reacher, where the action space is only 2-dimensional, policy-side baselines (GRAD $0.046$, CAPS/ASAP $0.048$) edged out PAVE ($sm$ $0.052$) by small margins, consistent with direct policy constraints being sufficient for simpler control manifolds. PAVE's distinct advantage emerged on higher-dimensional locomotion: Walker ($sm$ $0.748 \to 0.584$, $re$ $4907 \to 4954$) and Hopper ($sm$ $0.773 \to 0.556$, $re$ $3481 \to 3489$) saw substantial smoothness reductions while preserving Base returns, whereas ASAP achieved better absolute smoothness but at substantial return cost (Walker $re$

4731, Hopper 3077, both well below Base). On Ant, the three top methods clustered tightly: GRAD ($re$ 5729, $sm$ 1.632), ASAP ($sm$ 1.404, $re$ 5707), and PAVE ($re$ 5706, $sm$ 1.604). PAVE's high-dimensional advantage stems from the more volatile curvature and frequent gradient spikes inherent in such state-action spaces, where critic-centric regularization smoothed the gradient field while preserving Base returns.

### 6.3. Qualitative Analysis of the Q-Gradient Field

We qualitatively assessed the learning signal stability by visualizing the mixed-partial Hessian norm $\|\nabla^2_{sa}Q\|$ in Figure 1 across base TD3 and other regularization methods. The base TD3 and contemporary policy-side regularizers exhibited a "jagged" landscape characterized by numerous sharp spikes, representing a highly unstable learning signal. These jagged landscapes demonstrated that conventional policy-side regularization failed to mitigate the underlying geometric irregularities. In contrast, PAVE effectively paved the gradient field, inducing a flat and consistent landscape. This visualization confirmed that PAVE successfully minimized the mixed-partial volatility ($M$) as intended by our theoretical objective. A detailed description of the visualization protocol and corresponding plots for all six benchmark

*Table 3.* Performance under varying observation noise levels $\sigma \in \{0.01, 0.05, 0.1\}$ on the SiLU-unified TD3 setting (5 seeds). We compare the total average return (*re*) and smoothness score (*sm*) across various TD3-based regularization methods. Standard deviations shown in parentheses.

| | LunarLander | | | | | | Walker | | | | | |
| --- | --- | --- | --- | --- | --- | --- | --- | --- | --- | --- | --- | --- |
| | 0.01 | | 0.05 | | 0.1 | | 0.01 | | 0.05 | | 0.1 | |
| Method | *re*↑ | *sm*↓ | *re*↑ | *sm*↓ | *re*↑ | *sm*↓ | *re*↑ | *sm*↓ | *re*↑ | *sm*↓ | *re*↑ | *sm*↓ |
| Base | 213.2 (44.0) | 1.930 (0.798) | 209.6 (40.2) | 1.968 (0.684) | 189.4 (20.3) | 2.501 (0.607) | 4837.0 (420.9) | 2.012 (0.191) | 4840.8 (407.2) | 2.160 (0.161) | 4687.9 (384.7) | 2.386 (0.133) |
| CAPS | 230.7 (25.5) | 0.908 (0.262) | 227.0 (31.3) | 1.173 (0.440) | 213.1 (16.5) | 1.387 (0.263) | 5146.4 (320.1) | 1.607 (0.231) | 5250.0 (212.4) | 1.747 (0.233) | 5224.0 (141.9) | 1.937 (0.233) |
| GRAD | 251.3 (17.5) | 0.755 (0.053) | 244.9 (11.5) | 1.080 (0.108) | 213.1 (26.3) | 1.664 (0.166) | 4949.7 (303.1) | 1.389 (0.169) | 4908.8 (313.7) | 1.547 (0.163) | 4917.4 (323.0) | 1.828 (0.153) |
| ASAP | 161.6 (114.6) | 2.074 (0.964) | 160.8 (110.8) | 2.196 (1.061) | 150.4 (105.6) | 2.589 (0.896) | 4664.8 (450.5) | 1.625 (0.206) | 4784.2 (446.2) | 1.806 (0.193) | 4798.1 (439.8) | 2.055 (0.155) |
| PAVE | **254.5** (12.1) | **0.633** (0.206) | **247.8** (18.3) | **0.733** (0.083) | **219.1** (24.1) | **1.257** (0.205) | **5569.2** (444.9) | **1.296** (0.162) | **5579.1** (435.4) | **1.520** (0.116) | **5537.1** (472.8) | **1.856** (0.066) |

*Table 4.* MPR proxy validation: ratio of finite-difference proxy to exact analytic Hessian norm (1.0 = exact match), and the induced $M_{\text{sup}}$ reduction over Base. Absolute $M_{\text{sup}}$ values appear in Table 5 and Appendix G.

| Env | MPR Ratio | $M_{\text{sup}}$ Reduction |
| --- | --- | --- |
| Pend. | 1.14 | 2.1× |
| Hop. | 1.04 | 9.5× |
| Reach. | 1.02 | — |
| Walker | 0.91 | 3.0× |
| Ant | 0.72 | 1.4× |
| Lunar | 0.67 | 6.6× |

*Table 5.* Empirical validation of $L \leq M/\mu$: PAVE simultaneously suppresses $M_{\text{sup}}$ and lifts the strict-concavity rate (TD3, SiLU-unified, 5 seeds × 10 episodes). Full per-method results in Appendix G.

| Env | $M_{\text{sup}}$↓ Base / PAVE | Neg.Def↑ Base / PAVE |
| --- | --- | --- |
| Lunar | 990 / **151** | 0.35 / **0.84** |
| Pend. | 447 / **210** | 0.47 / **1.00** |
| Reach. | **16** / 26 | **1.00** / 0.99 |
| Ant | 8067 / **5962** | 0.17 / **0.66** |
| Hop. | 8831 / **934** | 0.26 / **0.86** |
| Walk. | 1923 / **643** | 0.14 / **0.64** |

environments are provided in Appendix A.

## 6.4. Proxy Validation

To quantitatively verify that the proxy losses approximated the intended Hessian quantities, we computed both the proxy (Eq. 11, finite-difference) and the exact analytic Hessian norm $\sigma^2 \|\nabla_{sa}^2 Q\|_F^2$ (Eq. 12, autograd) on the same trained PAVE critic under the SiLU-unified setting at each timestep (6 environments, 5 seeds, 10 episodes). Additionally, we measured $M_{\text{sup}} = \sup \|\nabla_{sa}^2 Q\|_2$ to verify end-to-end effectiveness. As summarized in Table 4: MPR ratios ranged from 0.67 to 1.14, confirming that the proxy approximated the Hessian norm. Despite the approximation gap, PAVE reduced $M_{\text{sup}}$ in 5/6 environments (up to 9.5×), validating that the proxy served its intended role as an optimization signal for reducing $M$. Full validation including VFC is provided in Appendix F.

## 6.5. Empirical Validation of the Theory

The bound $L \leq M/\mu$ predicts that controlling *both* terms is necessary: reducing $M$ alone is insufficient if $\mu$ collapses. We verified this on TD3 along three independent angles, summarized jointly in Table 5. (i) PAVE directly suppressed $M$: $M_{\text{sup}} = \sup \|\nabla_{sa}^2 Q\|_2$ dropped substantially

*Table 6.* Q-gradient temporal consistency on trained TD3 critics. cos↑ = mean cosine similarity between consecutive $\nabla_a Q$; flip↓ = fraction of timesteps with cos < 0.

| Env | Base cos↑ | PAVE cos↑ | Base flip↓ | PAVE flip↓ |
| --- | --- | --- | --- | --- |
| Lunar | 0.593 | **0.901** | 0.184 | **0.033** |
| Pend. | 0.491 | **0.954** | 0.254 | **0.023** |
| Reach. | 0.950 | **0.968** | 0.016 | **0.010** |
| Ant | 0.153 | **0.183** | 0.355 | **0.333** |
| Hop. | 0.566 | **0.755** | 0.165 | **0.073** |
| Walk. | 0.635 | **0.835** | 0.090 | **0.014** |

over Base in 5/6 environments (Table 5, $M_{\text{sup}}$ column; full per-method comparison in Appendix G). (ii) PAVE preserved $\mu$: the strict-concavity rate (fraction of $(s, a)$ at which $\lambda_{\max}(\nabla_{aa}^2 Q) < 0$) rose from 14–47% under unregularized critics to 64–100% under PAVE (Table 5, Neg.Def column), and this gap held *from early training onwards*, not merely at convergence (Appendix H). (iii) The action-gradient field became temporally consistent: PAVE substantially suppressed the cosine flip rate of $\nabla_a Q$ along rollouts (Sec. 6.6). A complementary failure-mode analysis against critic-side baselines (Spectral Normalization, Gradient Penalty) is deferred to Appendix J.

*Table 7.* Component analysis of each loss component in PAVE. We incrementally add the loss terms to the base TD3 algorithm to demonstrate their individual and combined effects.

| Configuration | LunarLander | | Walker | |
| --- | --- | --- | --- | --- |
| | re ↑ | sm ↓ | re ↑ | sm ↓ |
| Base | 227.8 (74.1) | 1.809 (1.311) | 4834.3 (423.0) | 1.990 (0.208) |
| $+\mathcal{L}_{\text{MPR}}$ | 223.6 (74.6) | 1.748 (1.187) | 4947.2 (486.7) | 1.841 (0.191) |
| $+\mathcal{L}_{\text{MPR}} + \mathcal{L}_{\text{VFC}}$ | 237.4 (58.7) | 1.177 (0.461) | 4648.2 (291.8) | 1.497 (0.241) |
| $+\mathcal{L}_{\text{MPR}} + \mathcal{L}_{\text{VFC}} + \mathcal{L}_{\text{Curv}}$ | **264.5** (24.9) | **0.541** (0.290) | **5562.9** (451.4) | **1.272** (0.172) |

### 6.6. Q-Gradient Direction Stability

A core hypothesis of PAVE is that the actor receives a geometrically unstable learning signal — the gradient direction $\nabla_a Q$ flips between adjacent timesteps. We measured the cosine similarity between consecutive Q-gradients $\nabla_a Q(s_t, a_t)$ and $\nabla_a Q(s_{t+1}, a_{t+1})$ on trained TD3 critics (5 seeds × 10 episodes); a negative cosine indicates a gradient direction flip. As summarized in Table 6, PAVE substantially lifted the cosine alignment and suppressed flip rates. PAVE reduced flip rates by 2–11× in 4/6 environments (Pendulum $25.4\% \rightarrow 2.3\%$, Walker $9.0\% \rightarrow 1.4\%$, Lunar $18.4\% \rightarrow 3.3\%$, Hopper $16.5\% \rightarrow 7.3\%$), directly evidencing the predicted stabilization of the Q-gradient field.

### 6.7. Evaluation under Noisy Observations

We evaluated performance under scale-aware observation noise defined as $\delta \sim \mathcal{U}(-\sigma, \sigma) \odot \sigma_{\text{base}}$. As summarized in Table 3, PAVE exhibited remarkable stability across varying noise levels. Crucially, PAVE demonstrated performance invariance to noise intensity; in the Walker task, the agent maintained nearly identical returns of approximately 5550 even as the noise magnitude increased tenfold from $\sigma = 0.01$ to $0.1$. This indicated that the paved Q-gradient field effectively decoupled input perturbations from action generation. Furthermore, while the smoothness score naturally increased with higher noise, PAVE exhibited a controlled and gradual degradation with scores ranging from 1.30 to 1.86 in the Walker task, whereas baseline methods often suffered from erratic control signal explosions or significant performance collapse under high-noise conditions.

### 6.8. Component Analysis

We analyzed the contribution of each component via an incremental ablation study. As shown in Table 7, the addition of $\mathcal{L}_{\text{MPR}}$ and $\mathcal{L}_{\text{VFC}}$ progressively improved policy smoothness. However, smoothness alone did not guarantee performance; notably, the Walker return dipped to 4648.2. Crucially, the integration of $\mathcal{L}_{\text{Curv}}$ was indispensable. It not

only recovered and boosted the return to 5562.9 but further enhanced smoothness. This result directly validated our theoretical bound $L \leq M/\mu$ by proving that minimizing volatility alone was insufficient if curvature vanished; by preventing the landscape from flattening, PAVE ensured the inverse Hessian did not act as an amplification factor for policy sensitivity. We provide a detailed sensitivity analysis, isolating the impact of each regularization coefficient, in Appendix B.

### 6.9. Complexity Analysis

While PAVE introduces geometric regularizers to stabilize the critic, it is designed to maintain linear computational complexity $\mathcal{O}(k + d)$, where $k$ and $d$ denote state and action dimensions, respectively. Unlike methods requiring explicit Hessian construction ($\mathcal{O}(d^2)$), PAVE utilizes finite difference approximations and Hutchinson's trace estimator to ensure scalability to high-dimensional tasks. A detailed complexity derivation is provided in Appendix C. Although Table 8 shows lower per-step throughput, this overhead does not scale proportionally to wall-clock convergence: under TD3, PAVE converges in ~1.0–1.4× the wall-clock time of Base (Appendix I). Since PAVE does not alter the actor architecture, inference speed remains identical to that of the base algorithm.

*Table 8.* Training Throughput (FPS) Comparison.

| Environment | BASE | CAPS | GRAD | ASAP | PAVE |
| --- | --- | --- | --- | --- | --- |
| LunarLander | 35.6 | 32.6 | 31.0 | 28.0 | 25.0 |
| Walker | 34.2 | 34.0 | 33.0 | 38.8 | 27.0 |

## 7. Limitations

Beyond the modest training overhead quantified in Sec. 6.9 (inference is unaffected since the actor is unchanged), the principal limitation is theoretical scope: the proxy losses incentivize—not guarantee—the geometric properties, and the strict concavity assumption holds in 64–100% of states under PAVE versus 14–47% under unregularized critics.

## 8. Conclusion

We identify the critic's irregular geometry as the root of policy instability and prove that policy sensitivity is governed by the ratio of the Q-function's mixed-partial volatility to its action-space curvature. PAVE stabilizes this geometry through critic-side losses, and our experiments confirm that paving the Q-gradient field smooths policies without any actor-side constraint, substantiating a critic-centric perspective on continuous control.

## Acknowledgments

This work was supported in part by the National Research Foundation of Korea (NRF) grant funded by the Korean government (MSIT) under Grant No. RS-2025-00564137, in part by Convergence security core talent training business support program under Grant IITP-2023-RS-2023-00266615, and in part by the Technology Innovation Program (RS-2025-25453780, Development of a National Humanoid AI Robot Foundation Model for Multi-Task Applications) funded by the Ministry of Trade Industry & Resources (MOTIR, Korea).

## Impact Statement

This paper presents work whose goal is to advance the field of machine learning. There are many potential societal consequences of our work, none of which we feel must be specifically highlighted here.

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

## A. Extended Analysis and Protocol for Q-Gradient Field Visualization

### A.1. Visualization Protocol

To qualitatively assess the geometric stability of the learning signal, we visualized the magnitude of the mixed-partial derivatives.

**Dominant Axis Selection.** For each environment, we first identified the pair of state dimension $s_i$ and action dimension $a_j$ that exhibited the strongest interaction in the vanilla model. This was defined by scanning all combinations and selecting the pair with the maximum mixed-partial derivative magnitude. By fixing the visualization of all compared algorithms to these "dominant axes" identified from the vanilla baseline, we ensure a fair comparison by focusing on the regions most susceptible to instability.

**Exact Computation.** Under the SiLU-unified setting, the critic is $C^2$-continuous, enabling exact Hessian computation via autograd. At each grid point, we constructed the full $d_a \times d_s$ mixed Hessian $\nabla_{sa}^2 Q$ by computing $\partial(\partial Q/\partial a_i)/\partial s$ for each action dimension, and extracted the spectral norm $\|\nabla_{sa}^2 Q\|_2$ via singular value decomposition. We swept the dominant state and action dimensions across a range of $[-1.0, 1.0]$ and $[-1.5, 1.5]$ respectively, centered at a reference state, on a $50 \times 50$ grid. For each environment, all methods share the same color scale (Z-axis clipped at the Base model's 99th percentile) to ensure fair visual comparison.

### A.2. Full Visualization Results Across Environments

Figure 2 and 3 presented the comparative 3D landscapes of the mixed-partial Hessian norm $\|\nabla_{sa}^2 Q\|$ for all six benchmark environments. This cross-environment evaluation confirmed that the geometric instability of the critic—characterized by high-volatility gradient fields—was a universal phenomenon in standard actor-critic training. Across all tested environments, including LunarLander, Pendulum, Reacher, Ant, Hopper, and Walker, the vanilla and existing policy-side regularizers exhibited noticeable landscape irregularities and a dense population of sharp gradient spikes. These spikes indicated that the steepest ascent direction rotated violently even with infinitesimal state shifts, and notably, current policy-side smoothness regularizers like ASAP failed to mitigate these underlying geometric singularities. In contrast, PAVE demonstrated a consistent ability to remove these erratic spikes and stabilize the landscapes into smooth manifolds across the entire suite of environments.

## B. Hyperparameter Sensitivity Analysis

Figure 4 illustrated the impact of varying each effective $\lambda$ on the Average Return and Smoothness Metric. At excessively high values, the smoothness metric notably deteriorated (dashed orange line increased), accompanied by a sharp decline in returns. This phenomenon directly validates our theoretical bound $L \leq M/\mu$. Minimizing gradient volatility ($M$) via MPR or VFC in isolation tends to flatten the Q-landscape, driving the action-space curvature ($\mu$, the denominator) toward zero. As the denominator vanishes, the norm of the inverse Hessian $\|\nabla_{aa}^2 Q\|^{-1}$ explodes, paradoxically amplifying policy sensitivity despite the regularization. In contrast, the curvature term ($\lambda_3$) demonstrated a critical role in stabilization by strictly lower-bounding $\mu$. It effectively prevented the vanishing gradient problem, ensuring that the inverse Hessian did not act as an amplification factor. The results highlighted that $\lambda_3$ acted as a necessary counterbalance, maintaining high returns and stability even when strong smoothness regularization was applied.

## C. Theoretical Complexity Analysis

To provide a rigorous assessment of PAVE's efficiency, we denote the dimensionality of the state space as $k = \dim(\mathcal{S})$ and the action space as $d = \dim(\mathcal{A})$. PAVE is engineered to impose second-order geometric constraints while maintaining linear computational scaling, $\mathcal{O}(k + d)$, avoiding the quadratic costs typically associated with explicit Hessian construction.

### C.1. Efficient Geometric Proxies

- **MPR & VFC**: Instead of computing the full mixed-partial Hessian $\nabla_{sa}^2 Q \in \mathbb{R}^{d \times k}$, which incurs a cost of $\mathcal{O}(kd)$, these terms utilize a finite-difference approximation grounded in a Taylor expansion. This requires only two gradient evaluations per update, effectively reducing the complexity to $\mathcal{O}(k + d)$.

- **Curvature Preservation**: The $\mathcal{L}_{\text{Curv}}$ term employs Hutchinson's trace estimator with random Rademacher vectors to

*Table 9.* Comparison of theoretical computational complexity per update step ($k$: state dim, $d$: action dim).

| Component | Standard Hessian-based | PAVE |
|---|---|---|
| Mixed-Partials ($\nabla^2_{sa}Q$) | $\mathcal{O}(kd)$ | $\mathcal{O}(k+d)$ |
| Action Curvature ($\nabla^2_{aa}Q$) | $\mathcal{O}(d^2)$ | $\mathcal{O}(d)$ |
| Actor Update | $\mathcal{O}(d)$ | $\mathcal{O}(d)$ |
| Total Complexity | $\mathcal{O}(kd+d^2)$ | $\mathcal{O}(k+d)$ |

*Table 10.* Detailed hyperparameters for each method in the Gymnasium-TD3 setting.

| Method | Gymnasium Environments | | | | | |
|---|---|---|---|---|---|---|
| | LunarLander | Pendulum | Reacher | Ant | Hopper | Walker |
| CAPS | $\lambda_T = 0.1$ $\lambda_S = 0.5$ $\sigma = 0.2$ | $\lambda_T = 1.0$ $\lambda_S = 5.0$ $\sigma = 0.2$ | $\lambda_T = 0.1$ $\lambda_S = 0.5$ $\sigma = 0.2$ | $\lambda_T = 0.1$ $\lambda_S = 0.5$ $\sigma = 0.2$ | $\lambda_T = 0.1$ $\lambda_S = 0.5$ $\sigma = 0.2$ | $\lambda_T = 0.1$ $\lambda_S = 0.5$ $\sigma = 0.2$ |
| GRAD | $\lambda_T = 1.0$ | $\lambda_T = 1.0$ | $\lambda_T = 1.0$ | $\lambda_T = 1.0$ | $\lambda_T = 1.0$ | $\lambda_T = 1.0$ |
| ASAP | $\lambda_T = 0.005$ $\lambda_S = 0.03$ $\lambda_P = 2.0$ | $\lambda_T = 0.005$ $\lambda_S = 0.03$ $\lambda_P = 2.0$ | $\lambda_T = 0.1$ $\lambda_S = 0.1$ $\lambda_P = 2.0$ | $\lambda_T = 0.05$ $\lambda_S = 0.3$ $\lambda_P = 2.0$ | $\lambda_T = 0.07$ $\lambda_S = 0.3$ $\lambda_P = 2.0$ | $\lambda_T = 0.05$ $\lambda_S = 0.3$ $\lambda_P = 2.0$ |
| PAVE | $\lambda_1 = 0.1$ $\lambda_2 = 0.1$ $\lambda_3 = 0.01$ | $\lambda_1 = 2.0$ $\lambda_2 = 0.005$ $\lambda_3 = 2.0$ | $\lambda_1 = 0.1$ $\lambda_2 = 0.1$ $\lambda_3 = 0.01$ | $\lambda_1 = 0.1$ $\lambda_2 = 0.005$ $\lambda_3 = 0.5$ | $\lambda_1 = 0.1$ $\lambda_2 = 0.005$ $\lambda_3 = 0.5$ | $\lambda_1 = 0.1$ $\lambda_2 = 0.1$ $\lambda_3 = 0.01$ |

calculate the Hessian-vector product (HVP). This stochastic estimation maintains $\mathcal{O}(d)$ complexity, avoiding the $\mathcal{O}(d^2)$ requirement for full entry-wise computation of $\nabla^2_{aa}Q$.

### C.2. Total Computational Overhead

As summarized in Table 9, PAVE maintains linear scaling across all components. Notably, these geometric regularizers are applied exclusively as auxiliary losses to the critic, leaving the actor update mechanism entirely unchanged. This decoupling ensures the framework's scalability for high-dimensional articulated control tasks such as the Walker and Ant benchmarks.

## D. Visualization of Learning Curves

Figures 5 and 6 presented the learning curves of across various Gymnasium environments. We reported the performance of our proposed method against several baselines including BASE, CAPS, GRAD, and ASAP. Each curve illustrated the mean episodic reward over five independent seeds, with shaded regions representing the standard deviation.

## E. Extended Implementation Details

### E.1. Report on Hyperparameter Setting

We provide the hyperparameters for each method in the Gymnasium setting in Table 10 and 11.

**PAVE hyperparameter selection.** The three loss weights $\lambda_1$ (MPR), $\lambda_2$ (VFC), and $\lambda_3$ (Curv) in Eq. 16 were selected via a coordinate search starting from the default $\lambda_1 = 0.1$, $\lambda_2 = 0.1$, $\lambda_3 = 0.01$, with each weight swept one at a time. The tuning order was $\lambda_3$ first (most sensitive, since it gates the curvature denominator $\mu$ in $L \leq M/\mu$), followed by $\lambda_2$, then $\lambda_1$. The perturbation scale $\sigma = 0.01$ for MPR and the curvature floor $\delta = 1.0$ for Curv were fixed across all environments. The resulting per-environment values reflect this coordinate search and are consistent with the sensitivity analysis reported in Appendix K.

*Table 11.* Detailed hyperparameters for each method in the Gymnasium-SAC setting.

| Method | Gymnasium Environments | | | | | |
| --- | --- | --- | --- | --- | --- | --- |
| | LunarLander | Pendulum | Reacher | Ant | Hopper | Walker |
| CAPS | $\lambda_T = 0.1$ $\lambda_S = 0.5$ $\sigma = 0.2$ | $\lambda_T = 1.0$ $\lambda_S = 5.0$ $\sigma = 0.2$ | $\lambda_T = 0.1$ $\lambda_S = 0.5$ $\sigma = 0.2$ | $\lambda_T = 0.1$ $\lambda_S = 0.5$ $\sigma = 0.2$ | $\lambda_T = 0.1$ $\lambda_S = 0.5$ $\sigma = 0.2$ | $\lambda_T = 0.1$ $\lambda_S = 0.5$ $\sigma = 0.2$ |
| GRAD | $\lambda_T = 1.0$ | $\lambda_T = 1.0$ | $\lambda_T = 1.0$ | $\lambda_T = 1.0$ | $\lambda_T = 1.0$ | $\lambda_T = 1.0$ |
| ASAP | $\lambda_T = 0.005$ $\lambda_S = 0.03$ $\lambda_P = 2.0$ | $\lambda_T = 0.005$ $\lambda_S = 0.03$ $\lambda_P = 2.0$ | $\lambda_T = 0.1$ $\lambda_S = 0.1$ $\lambda_P = 2.0$ | $\lambda_T = 0.05$ $\lambda_S = 0.3$ $\lambda_P = 2.0$ | $\lambda_T = 0.07$ $\lambda_S = 0.3$ $\lambda_P = 2.0$ | $\lambda_T = 0.05$ $\lambda_S = 0.3$ $\lambda_P = 2.0$ |
| PAVE | $\lambda_1 = 0.1$ $\lambda_2 = 0.5$ $\lambda_3 = 0.05$ | $\lambda_1 = 0.1$ $\lambda_2 = 0.005$ $\lambda_3 = 0.5$ | $\lambda_1 = 0.1$ $\lambda_2 = 5\text{e}{-}4$ $\lambda_3 = 1.0$ | $\lambda_1 = 0.1$ $\lambda_2 = 5\text{e}{-}4$ $\lambda_3 = 1.0$ | $\lambda_1 = 2.0$ $\lambda_2 = 5\text{e}{-}4$ $\lambda_3 = 3.0$ | $\lambda_1 = 2.0$ $\lambda_2 = 0.005$ $\lambda_3 = 2.0$ |

## E.2. Hardware Setting

We utilized distinct computational resources depending on the experimental phase, while ensuring that all comparative algorithms within each benchmark were evaluated on identical hardware settings to guarantee fair comparison. The primary training for all TD3-based methods (including performance and component analysis) was conducted on a server equipped with an Intel Xeon Gold 6226R CPU (@ 2.90GHz) and an NVIDIA GeForce RTX 3090 GPU. Similarly, all SAC-based experiments were executed on a workstation featuring an Intel Xeon Gold 6426Y CPU and an NVIDIA RTX A4000 GPU. In contrast, the robustness analysis and Q-gradient field visualization were performed using pretrained models on a separate machine with an AMD Ryzen 7 5800X 8-Core Processor.

## F. Proxy Validation

We validated the approximation quality of MPR (Eq. 11 finite-difference $\approx$ Eq. 12 exact analytic) and VFC (Eq. 13 finite-difference $\approx$ Eq. 14 exact analytic) by computing both quantities on the same trained PAVE critic under the SiLU-unified setting (6 environments $\times$ 5 seeds $\times$ 10 episodes; analytic side via autograd-based Hessian construction).

**MPR Validation.** As derived in Eq. 12, $\mathcal{L}_{\text{MPR}} \approx \sigma^2 \|\nabla_{sa}^2 Q\|_F^2$. We computed both sides at each timestep; Table 12 reports the ratio.

*Table 12.* MPR proxy validation: finite-difference $\mathcal{L}_{\text{MPR}}$ (Eq. 11) vs the exact analytic Hessian norm $\sigma^2 \|\nabla_{sa}^2 Q\|_F^2$ (Eq. 12) on trained PAVE critics (SiLU-unified, 5 seeds $\times$ 10 episodes). Ratio = Proxy / Analytic.

| Env | Proxy (Eq. 11) | $\sigma^2 \|\nabla_{sa}^2 Q\|_F^2$ (Eq. 12) | Ratio |
| --- | --- | --- | --- |
| Pend. | 0.054 | 0.048 | 1.14 |
| Hop. | 0.516 | 0.496 | 1.04 |
| Reach. | 0.007 | 0.007 | 1.02 |
| Walker | 0.473 | 0.522 | 0.91 |
| Ant | 5.109 | 7.097 | 0.72 |
| Lunar | 0.137 | 0.205 | 0.67 |

In 4/6 environments the ratio lies within 0.91–1.14. The remaining two show ratio $< 1$, where the proxy underestimates due to higher-order terms — a conservative direction for optimization.

**VFC Validation.** VFC used the same finite-difference structure with actual transitions $\Delta s$ instead of random $\varepsilon$. The Taylor approximation degraded for large $\|\Delta s\|$, as quantified in Table 13.

*Table 13.* VFC proxy validation: finite-difference $\mathcal{L}_{\text{VFC}}$ (Eq. 13) vs the directional Hessian norm $\|\nabla^2_{sa}Q \cdot \Delta s\|^2$ (Eq. 14) on trained PAVE critics.

| Env | Proxy (Eq. 13) | $\|\nabla^2_{sa}Q \cdot \Delta s\|^2$ (Eq. 14) | Ratio |
|---|---|---|---|
| Pend. | 1.053 | 1.152 | 0.91 |
| Reach. | 0.001 | 0.001 | 0.77 |
| Walker | 1.930 | 6.137 | 0.31 |
| Hop. | 0.654 | 2.170 | 0.30 |
| Lunar | 0.093 | 0.947 | 0.10 |
| Ant | 104.3 | 3087.0 | 0.03 |

VFC ratios spanned a wide range from 0.03 (Ant) to 0.91 (Pendulum), reflecting that the first-order Taylor approximation degraded for high-dimensional or discontinuous transitions. The proxies are designed to provide an optimization signal that reduces $M$, not to replicate the Hessian exactly. Despite the approximation gap, PAVE reduced $M_{\text{sup}}$ in 5/6 environments (see below).

## G. Exact Hessian Measurements

### G.1. $M_{\text{sup}}$ (Spectral Norm of Mixed Hessian)

We measured $M = \sup \|\nabla^2_{sa}Q\|_2$ by constructing the full $d_a \times d_s$ mixed Hessian via per-action-dimension autograd and extracting the largest singular value at each timestep (5 seeds $\times$ 10 episodes, trajectory-wise maximum). Table 14 reports the results across the six environments.

*Table 14.* $M_{\text{sup}} = \sup \|\nabla^2_{sa}Q\|_2$ on trained TD3 critics (SiLU-unified, 5 seeds $\times$ 10 episodes). PAVE achieves the lowest $M_{\text{sup}}$ in 5/6 environments.

| | Lunar | Pend. | Reach. | Ant | Hop. | Walk. |
|---|---|---|---|---|---|---|
| Base | 990 | 447 | **16** | 8067 | 8831 | 1923 |
| CAPS | 1218 | 431 | 24 | 6452 | 1070 | 1624 |
| GRAD | 1273 | 338 | 18 | 7920 | 2479 | 1990 |
| ASAP | 1227 | 393 | 17 | 7964 | 27627 | 1772 |
| **PAVE** | **151** | **210** | 26 | **5962** | **934** | **643** |

PAVE achieved the lowest $M_{\text{sup}}$ in 5/6 environments.

### G.2. Negative Definiteness Rate

We computed the full eigenvalue decomposition of $\nabla^2_{aa}Q$ via `eigvalsh` after symmetrization. Negative definiteness rate = fraction of $(s, a)$ where all eigenvalues $< 0$; Table 15 summarizes the results.

*Table 15.* Negative-definiteness rate of $\nabla^2_{aa}Q$ on trained TD3 critics (SiLU-unified, 5 seeds $\times$ 10 episodes). Fraction of $(s, a)$ where all eigenvalues $< 0$. PAVE raises strict concavity by 2.1–4.5$\times$ over Base in 5/6 environments; policy-side regularizers (CAPS/GRAD/ASAP) stay close to Base across most environments.

| Method | Lunar | Pend. | Reach. | Ant | Hop. | Walk. |
|---|---|---|---|---|---|---|
| Base | 0.352 | 0.466 | **1.000** | 0.174 | 0.259 | 0.141 |
| CAPS | 0.284 | 0.381 | 0.999 | 0.175 | 0.252 | 0.160 |
| GRAD | 0.389 | 0.697 | **1.000** | 0.196 | 0.185 | 0.161 |
| ASAP | 0.369 | 0.770 | **1.000** | 0.178 | 0.302 | 0.132 |
| **PAVE** | **0.842** | **0.995** | 0.994 | **0.659** | **0.856** | **0.637** |

PAVE improved concavity satisfaction in 5/6 environments (2.1–4.5$\times$ over Base). Notably, the three policy-side regularizers (CAPS, GRAD, ASAP) yielded rates close to the unregularized Base critic across most environments (with Pendulum as a partial exception where GRAD/ASAP rose to 0.70–0.77)—confirming that actor-side regularization largely leaves the critic's curvature geometry untouched, while only PAVE's critic-side $\mathcal{L}_{\text{Curv}}$ raises strict concavity into the 0.6–1.0 regime.

## H. Geometry Evolves During Training

The bound $L \leq M/\mu$ also makes a temporal prediction: as TD-learning fits a more complex Q-surface, $\|\nabla_{sa}^2 Q\|$ should grow over training unless an active constraint counters it. We checkpointed the critic at intermediate steps (SiLU-unified TD3) and recomputed $M_{\text{sup}}$ together with the negative-definiteness rate. Tables 16 and 17 report the trajectory on LunarLander and Walker.

*Table 16.* $M_{\text{sup}}$ and negative-definiteness rate at intermediate training checkpoints on LunarLander (TD3, SiLU-unified).

| Step | | $M_{\text{sup}} \downarrow$ | Neg.Def Rate↑ |
|------|------|------|------|
| 100K | Base / PAVE | 273 / **101** | 0.277 / **0.497** |
| 300K | Base / PAVE | 616 / **129** | 0.283 / **0.689** |
| 500K | Base / PAVE | 1105 / **188** | 0.332 / **0.808** |

*Table 17.* $M_{\text{sup}}$ and negative-definiteness rate at intermediate training checkpoints on Walker (TD3, SiLU-unified).

| Step | | $M_{\text{sup}} \downarrow$ | Neg.Def Rate↑ |
|------|------|------|------|
| 200K | Base / PAVE | 636 / **167** | 0.071 / **0.548** |
| 600K | Base / PAVE | 1212 / **417** | 0.121 / **0.600** |
| 1M | Base / PAVE | 1320 / **659** | 0.144 / **0.638** |

Base $M_{\text{sup}}$ grew roughly $4\times$ over Lunar training ($273 \rightarrow 1105$) and stayed high on Walker, confirming the predicted drift of the critic's mixed-partial volatility. PAVE, in contrast, kept $M_{\text{sup}}$ near constant ($<2\times$ growth on Lunar) and steadily lifted the strict-concavity rate from $<33\%$ to 64–81%. Critic-side regularization therefore counteracted the geometric pathology *from early training onwards*—not merely at convergence—directly supporting the active role of $\mathcal{L}_{\text{Curv}}$ and the proxy losses across the training trajectory.

## I. Wall-Clock Convergence Time

We measured wall-clock convergence time from the same SiLU-unified TD3 runs reported in Table 1. Convergence is defined as the first time the smoothed learning curve (uniform filter, window=10) reaches the method's own final performance (mean of the last 10% of training). Table 18 reports the per-environment convergence times.

*Table 18.* Wall-clock convergence time on TD3 (SiLU-unified). Convergence = first time the smoothed learning curve (uniform filter, window=10) reaches the method's own final performance (last 10% mean). 5 paper seeds, measured from TensorBoard logs.

| TD3 | Lunar | Pend. | Reach. | Ant | Hop. | Walk. |
|------|------|------|------|------|------|------|
| Base (min) | 206 | 26 | 62 | 392 | 202 | 400 |
| PAVE (min) | 203 | 36 | 79 | 453 | 212 | 514 |
| ratio | **0.99**× | 1.38× | 1.27× | 1.16× | 1.05× | 1.29× |

Under SiLU-unified conditions, TD3 convergence ratios ranged from 0.99–1.38×, with PAVE essentially tied with Base on Lunar (0.99×). Inference speed is identical since PAVE does not modify the actor architecture.

## J. Critic-Centric Baseline Comparison

We compared against two critic-side baselines that act on $M$-related quantities: SN-Critic, which applies spectral normalization to the critic following the RL adaptation of (Gogianu et al., 2021; Bjorck et al., 2021), and GP-Critic, which adds a gradient penalty $\lambda \|\nabla_a Q\|^2$ on the critic in the spirit of (Roth et al., 2017; Sokolić et al., 2017). Both were trained under TD3 with SiLU on Lunar (L) and Walker (W), with results in Table 19.

*Table 19.* Critic-side baseline comparison on TD3 (SiLU-unified) for Lunar (L) and Walker (W). SN-Critic collapses entirely; GP-Critic reduces $M$ modestly but leaves $\mu$ at Base. Only PAVE moves both numerator and denominator of the bound.

| Method | sm↓ (L/W) | re↑ (L/W) | $M_{\text{sup}}$↓ (L/W) | Neg Def↑ (L/W) |
|---|---|---|---|---|
| Base | 1.81 / 1.99 | 228 / 4834 | 990 / 1923 | 0.35 / 0.14 |
| SN-Critic | 0.00 / 0.00 | −970 / 14 | 0.1 / – | 0.40 / 0.18 |
| GP-Critic | 2.18 / 1.66 | 202 / 5011 | 882 / 1090 | 0.37 / 0.15 |
| **PAVE** | **0.54 / 1.27** | **264 / 5563** | **151 / 643** | **0.84 / 0.64** |

SN-Critic suppressed both $M$ and $\mu$ indiscriminately and collapsed the policy ($re = -970$ on Lunar, $14$ on Walker). GP-Critic reduced $M_{\text{sup}}$ modestly yet left $\mu$ at Base level—never beating Base smoothness on Lunar and only marginally improving it on Walker, far above PAVE's. Only PAVE—whose three losses target $\nabla^2_{sa}Q$ and $|\text{tr}\,\nabla^2_{aa}Q|$ on independent axes—moved both the numerator and the denominator of the bound in the right direction, exactly as the theory predicts.

**GP-Critic per-$\lambda$ results.** We swept the gradient penalty weight $\lambda$ across multiple orders of magnitude to verify that the failure was structural rather than a tuning artifact (Table 20). No setting beat Base smoothness on Lunar; on Walker, only the smallest weights retained task performance, but smoothness remained far above PAVE's.

*Table 20.* GP-Critic gradient-penalty weight $\lambda$ sweep on Lunar and Walker. No setting beats Base smoothness on Lunar; on Walker only small weights retain task performance, yet smoothness remains above PAVE's.

| $\lambda$ | Lunar | | Walker | |
|---|---|---|---|---|
| | $sm \downarrow$ | $re \uparrow$ | $sm \downarrow$ | $re \uparrow$ |
| 0.01 | 2.12 | **196** | **1.66** | **5012** |
| 0.1 | **2.18** | 203 | 1.96 | 4990 |
| 1.0 | 3.30 | 180 | 2.28 | 4142 |
| 10.0 | — | — | 0.49 | 484 (collapse) |
| PAVE | **0.54** | **264** | **1.27** | **5563** |

# K. Hyperparameter Sensitivity

Sensitivity sweep on Lunar and Walker (seeds #6–10, different hardware from main experiments). Default: $\lambda_1 = 0.1$, $\lambda_2 = 0.1$, $\lambda_3 = 0.01$. Tables 21 and 22 report smoothness and return sensitivities respectively.

*Table 21.* Smoothness sensitivity (sm↓) to PAVE hyperparameters on Lunar / Walker. Scale factors ×0.01, ×1, ×100 applied to default values ($\lambda_1 = 0.1$, $\lambda_2 = 0.1$, $\lambda_3 = 0.01$).

| Scale | $\lambda_3$ (Curv) | $\lambda_2$ (VFC) | $\lambda_1$ (MPR) |
|---|---|---|---|
| ×0.01 | 1.35 / 1.78 | 1.13 / 1.70 | 0.63 / 1.61 |
| ×1 | **0.54 / 1.52** | **0.54 / 1.52** | **0.54 / 1.52** |
| ×100 | 0.32 / 1.31 | 0.47 / 1.27 | 0.60 / 1.80 |

*Table 22.* Return sensitivity (re↑) to PAVE hyperparameters on Lunar / Walker. Same scale factors as Table 21.

| Scale | $\lambda_3$ re | $\lambda_2$ re | $\lambda_1$ re |
|---|---|---|---|
| ×0.01 | 250 / 4753 | 156 / 5142 | 262 / 4728 |
| ×1 | **255 / 5383** | **255 / 5383** | **255 / 5383** |
| ×100 | 261 / 4253 | 264 / 4480 | 257 / 4598 |

$\lambda_3$ (Curv) was the most sensitive parameter; $\lambda_1$ (MPR) was the most stable.

## L. Full Factorial Component Ablation

To complement the incremental ablation in Table 7, we ran the full $2^3$ factorial over the three PAVE components on Walker (TD3, SiLU-unified, conducted on different hardware from the main experiments). This isolates the marginal contribution of each loss and the interactions among them; results are in Table 23.

*Table 23.* Full $2^3$ factorial ablation over the three PAVE components on Walker (TD3, SiLU-unified, different hardware from main experiments). $\mathcal{L}_{\text{Curv}}$ is the prerequisite: without it, MPR+VFC worsens smoothness beyond Base.

| Configuration | Curv | Walker re↑ | Walker sm↓ |
|---|---|---|---|
| Base | – | 4589 | 1.828 |
| MPR only | × | 5059 | 1.754 |
| VFC only | × | 4788 | 1.823 |
| MPR + VFC | × | 5010 | 2.095 |
| Curv only | ✓ | 4896 | 1.650 |
| MPR + Curv | ✓ | 5177 | 1.718 |
| VFC + Curv | ✓ | 5537 | 1.918 |
| **MPR + VFC + Curv (Full)** | ✓ | **5502** | **1.483** |

Two findings stood out. First, $\mathcal{L}_{\text{Curv}}$ was a prerequisite: without it, the MPR+VFC combination actually *worsened* smoothness beyond Base (sm $2.095 > 1.828$), directly reflecting Prop. 4.3 — reducing $M$ alone is insufficient if $\mu$ is not preserved. Second, given $\mathcal{L}_{\text{Curv}}$, MPR and VFC contributed distinct value: MPR+Curv yielded the best 2-term smoothness (1.718) while VFC+Curv yielded higher return (5537) but worse smoothness (1.918). Only the full combination achieved both the best sm (1.483) and competitive re (5502).

## M. Combination with Actor-Side Regularization

To examine whether actor-side regularization provides additional benefit on top of PAVE's critic-side stabilization, we combined PAVE with CAPS (Mysore et al., 2021), a representative actor-side smoothness regularizer. All four methods share the SiLU-unified TD3 backbone on LunarLander; Table 24 reports the comparison.

*Table 24.* Combination with actor-side regularization (CAPS) on LunarLander (TD3, SiLU-unified). CAPS+PAVE performs comparably to PAVE alone, supporting critic-side sufficiency.

| Method | sm↓ | re↑ |
|---|---|---|
| Base | 1.809 | 227.8 |
| CAPS (actor-side) | 0.702 | 249.0 |
| PAVE (critic-side) | **0.541** | **264.5** |
| CAPS + PAVE | 0.543 | 264.0 |

CAPS alone improved smoothness substantially over Base (sm $1.809 \rightarrow 0.702$). PAVE further improved both sm (0.541) and re (264.5). When combined, CAPS+PAVE (sm 0.543, re 264.0) performed comparably to PAVE alone, suggesting that once the critic geometry is stabilized, the actor naturally tracks the smooth implied policy without additional output-level constraints. This empirically supported the central claim of the paper: regularizing the critic geometry is a sufficient route to policy smoothness in standard actor-critic methods.

## N. Extension to Humanoid

We extended the evaluation to *Humanoid-v5* under the SiLU-unified TD3 setting. This environment was initially excluded from the main experiments because vanilla TD3 failed to learn meaningfully here, rendering the smoothness metric uninformative (a near-zero policy trivially yielded a very low sm score), as shown in Table 25.

*Table 25.* Extension to Humanoid-v5 (TD3, SiLU-unified). PAVE enables non-trivial learning ($309 \rightarrow 2894$); the accompanying sm rise reflects Base TD3's degenerate near-constant policy.

| Method | re↑ | sm↓ |
|---|---|---|
| Base TD3 | 309 | 0.06 |
| **PAVE** | **2894** | 0.76 |

PAVE enabled TD3 to learn a non-trivial policy in Humanoid (re $309 \rightarrow 2894$, $9.4\times$ improvement). The accompanying rise in sm ($0.06 \rightarrow 0.76$) reflected the fact that Base TD3 produced near-constant outputs — a degenerate "smoothness" that disappeared once the agent actually controlled the body. Comparable smoothness numbers in Tables 1–2 should be read against the corresponding return; PAVE's Humanoid sm of $0.76$ was competitive with the values it achieved on the locomotion suite (Hopper $0.95$, Walker $1.27$) while delivering meaningful task performance.

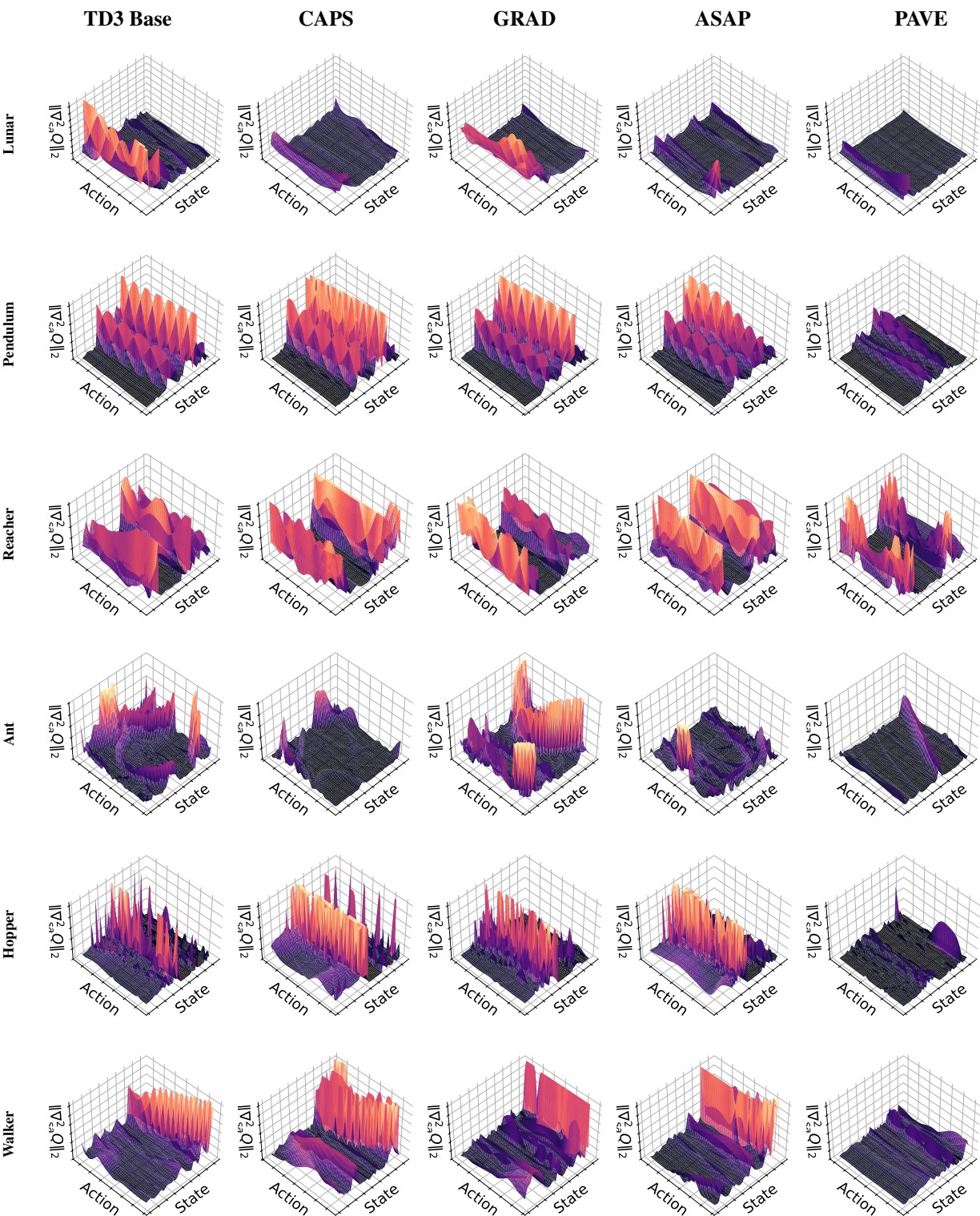

*Figure 2.* Comprehensive 3D visualization of the mixed-partial Hessian norm $\|\nabla_{sa}^2 Q\|$ across six Gymnasium environments. Each row corresponds to an environment, and each column represents a different stabilization method. While baseline methods exhibit highly irregular landscapes with sharp spikes (indicating an unstable learning signal), PAVE effectively paves the Q-gradient field, providing a smooth and stable landscape. For each environment, the Z-axis is clipped at the Base model's 99th percentile for consistent comparison across methods.

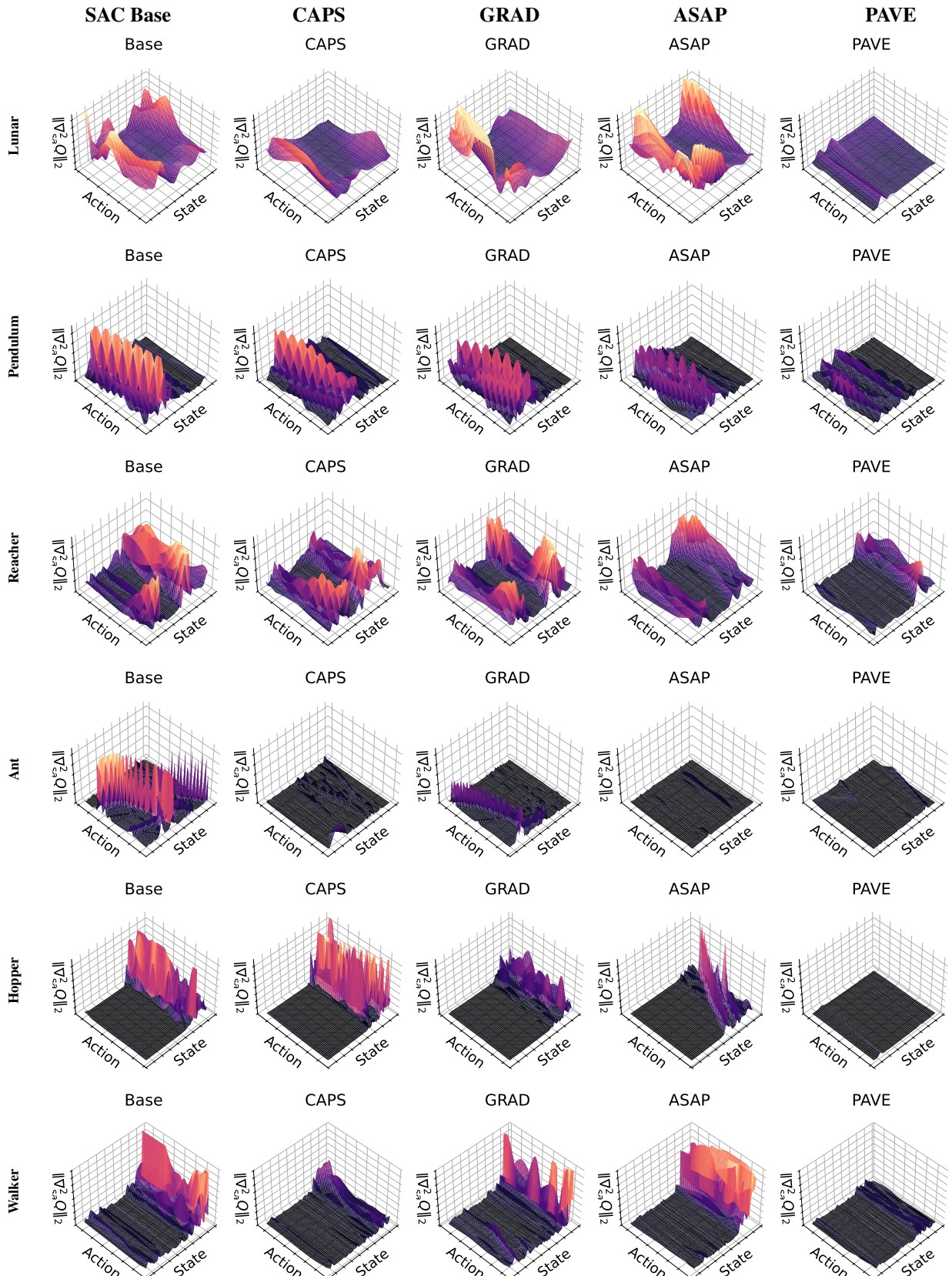

*Figure 3.* Comprehensive 3D visualization of the mixed-partial Hessian norm $\|\nabla_{sa}^2 Q\|$ across six Gymnasium environments. Each row corresponds to an environment, and each column represents a different stabilization method. While baseline methods exhibit highly irregular landscapes with sharp spikes (indicating an unstable learning signal), PAVE effectively paves the Q-gradient field, providing a smooth and stable landscape. For each environment, the Z-axis is clipped at the Base model's 99th percentile for consistent comparison across methods.

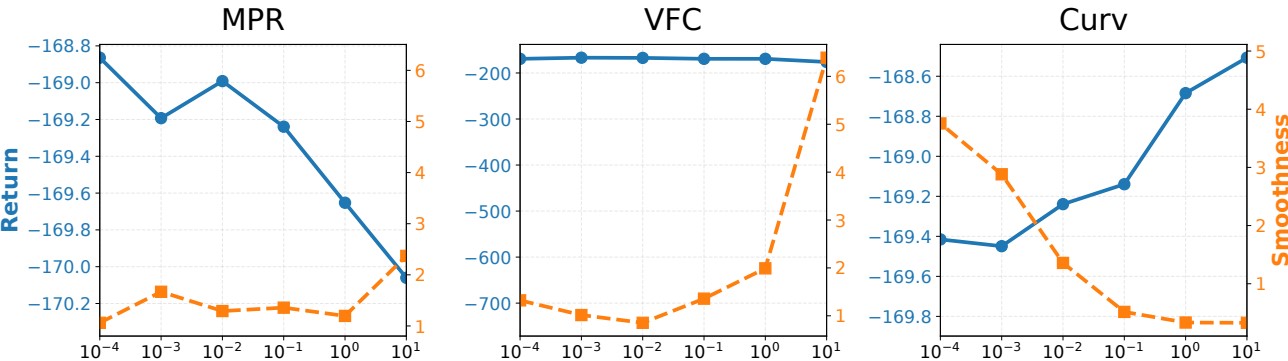

*Figure 4.* Hyperparameter Sensitivity on Pendulum. For each subplot, only the target parameter was varied while the other two were held fixed at their default settings.

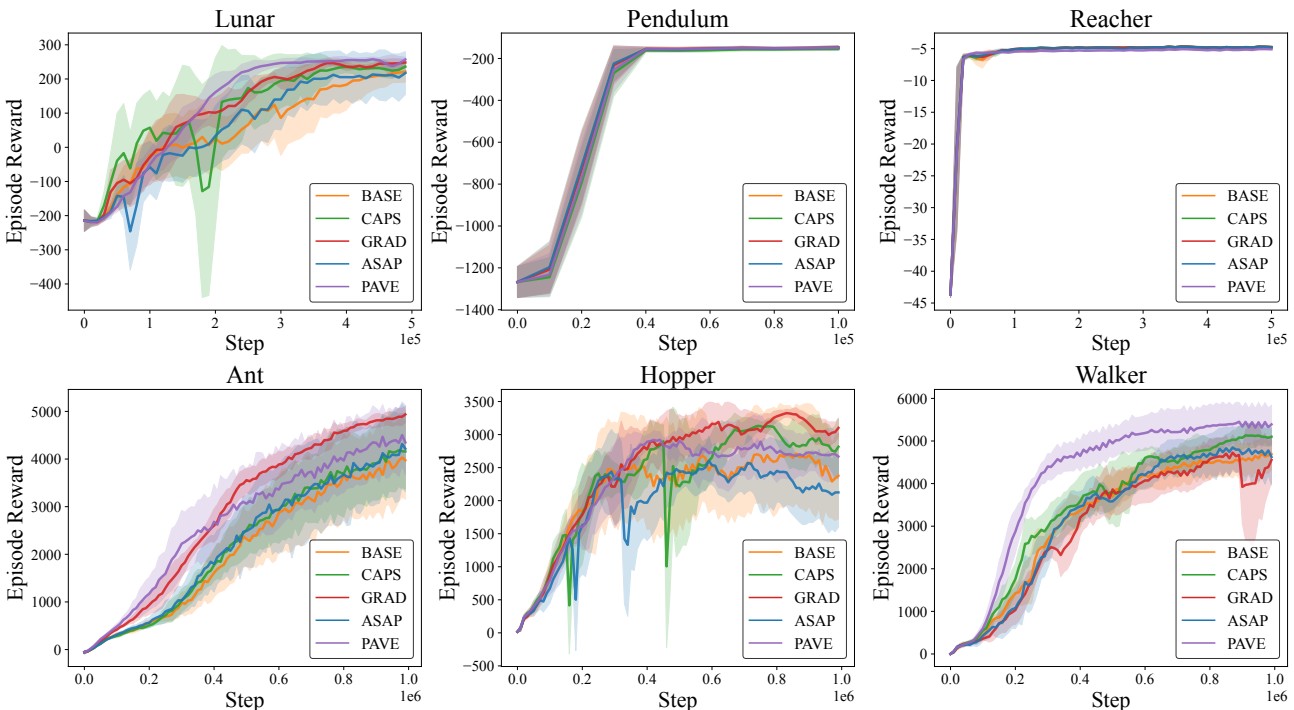

*Figure 5.* Learning curves of the TD3 algorithm across various Gymnasium environments.

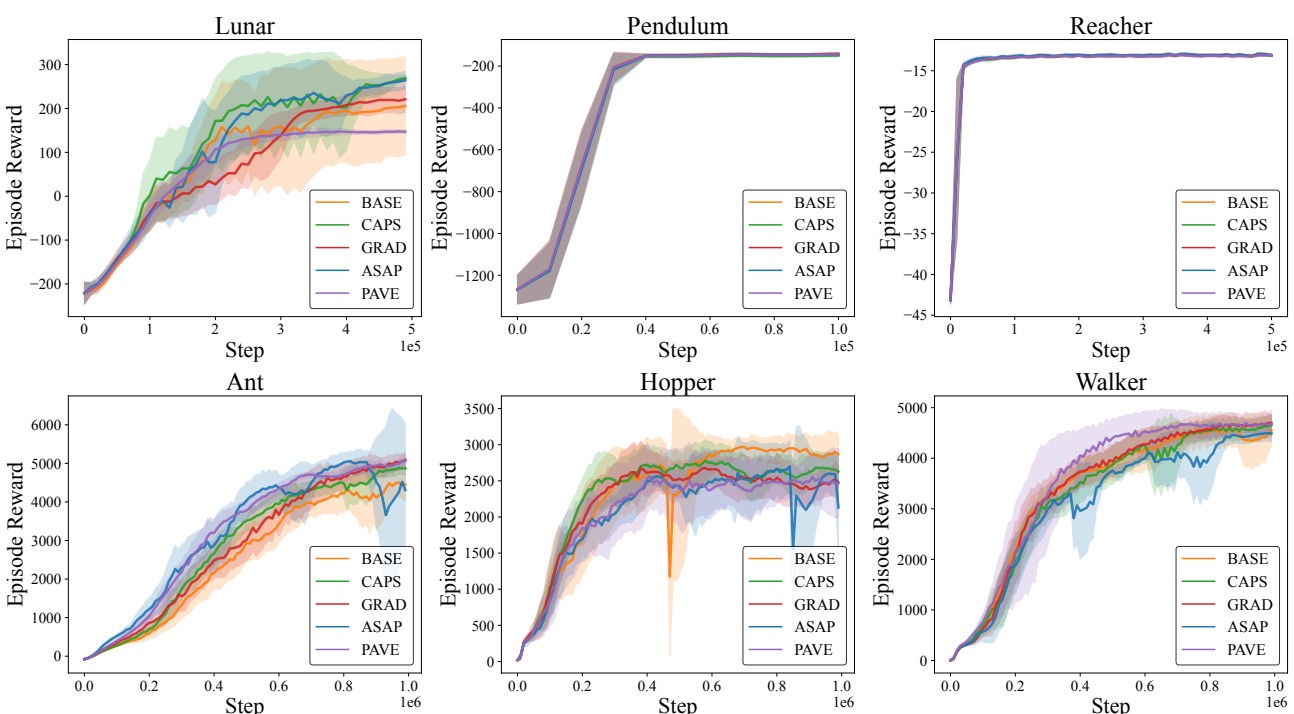

*Figure 6.* Learning curves of the SAC algorithm across various Gymnasium environments.

