# OpenReview forum: "Stabilizing the Q-Gradient Field for Policy Smoothness in Actor-Critic Methods"
_ICML.cc/2026/Conference — ICML 2026 spotlight_

### Official Review · Reviewer_cvdZ · 2026-03-03

**Soundness:** 4
**Presentation:** 4
**Significance:** 3
**Originality:** 3
**Overall Recommendation:** 5
**Confidence:** 3

**Summary:**

This paper studies the problem that actor–critic policies often exhibit high-frequency actions, which hurts physical deployment. The central issue is how to obtain policy smoothness without directly regularizing the actor outputs. The paper argues that non-smoothness is fundamentally governed by the critic’s differential geometry. Motivated by this, the authors propose Policy-Aware Value-field Equalization (PAVE). Experiments on Gymnasium/MuJoCo with TD3 and SAC report improved smoothness while maintaining competitive returns.

**Compliance With Llm Reviewing Policy:**

Affirmed.

**Final Justification:**

The authors’ rebuttal has adequately addressed my concerns, so I will maintain my positive score.

**Key Questions For Authors:**

I follow most of the parts so I only have two questions.

1. Why do we need both MPR and VFC? MPR and VFC both seem aimed at reducing the M term by stabilizing changes of Q across nearby states. Could you clarify the distinct role each plays?

2. PAVE only regularizes the critic and suggests the implied greedy policy becomes less sensitive. However, the final deployed policy is produced by an actor network trained with policy gradients; if the actor fails to accurately track the implied smooth policy, the output may still be sensitive. Are there complementary actor-side methods to reduce sensitivity? Is critic-side regularization alone sufficient?

**Limitations:**

No, the authors did not discuss limitations in the paper.

**Strengths And Weaknesses:**

The paper is easy to follow: it starts from the formulation of policy sensitivity, identifies two key Hessian-related contributors, then proposes PAVE and supports it with experiments. The authors focus on a relevant challenge in continuous control: reducing oscillatory actions without sacrificing task performance. To my best knowledge, the methods used in the paper is a fresh perspective.

---

> ### Author Rebuttal · Authors · 2026-03-30
>
> We appreciate the reviewer for the careful reading and for recognizing PAVE as a "fresh perspective." We address each question below.
>
> **S1. Distinct Roles of MPR and VFC**
>
> We thank the reviewer for this thoughtful question. Both MPR and VFC target M (numerator of L ≤ M/μ), but they cover complementary directions in state space:
>
> - **MPR** perturbs states isotropically (s + ε, ε ~ N(0, σ²I)), suppressing Q-gradient sensitivity in all directions.
> - **VFC** operates along actual transition directions (s_t → s_{t+1}), targeting the gradient changes the policy encounters during execution.
>
> To isolate their contributions, we ran a full 2³ factorial ablation on Walker (SiLU, re-run on different hardware from the submitted manuscript):
>
> | Config | Curv | Walker re↑ | Walker sm↓ |
> |---|---|---|---|
> | Base | | 4589 | 1.828 |
> | MPR only | × | 5059 | 1.754 |
> | VFC only | × | 4788 | 1.823 |
> | MPR+VFC | × | 5010 | 2.095 |
> | Curv only | ✓ | 4896 | 1.650 |
> | MPR+Curv | ✓ | 5177 | 1.718 |
> | VFC+Curv | ✓ | 5537 | 1.918 |
> | **MPR+VFC+Curv** | ✓ | 5502 | **1.483** |
>
> L_Curv is a prerequisite: without it, MPR+VFC worsens smoothness beyond Base (sm 2.095 > 1.828), directly reflecting Prop. 4.2: reducing M alone is insufficient if μ is not preserved. Given L_Curv, both components contribute distinct value: MPR+Curv achieves better sm (1.718), while VFC+Curv achieves higher re (5537) but worsens sm (1.918). Only the full combination achieves both the best sm (1.483) and competitive re.
>
> **S2. Sufficiency of Critic-Side Regularization**
>
> We appreciate the reviewer raising this point. As the reviewer correctly notes, the final policy is produced by the actor network. The key insight is that both deterministic (TD3) and stochastic (SAC) policy updates are fundamentally driven by ∇_a Q from the critic (Sec. 3). If this learning signal points in inconsistent directions for neighboring states, the actor receives contradictory updates and learns an erratic policy. PAVE addresses this root cause by stabilizing ∇_a Q across states (MPR/VFC) and preserving curvature (L_Curv). This is the practical implication of Lemma 4.1: policy sensitivity is determined by critic geometry.
>
> Empirically, PAVE achieves the best smoothness in all 6 TD3 environments without any actor modification. To directly examine whether actor-side regularization provides additional benefit, we combined PAVE with CAPS (Mysore et al., 2021), a representative actor-side smoothness regularizer. On Lunar (TD3, SiLU):
>
> | Method | sm↓ | re↑ |
> |---|---|---|
> | Base | 1.809 | 227.8 |
> | CAPS (actor) | 0.702 | 249.0 |
> | **PAVE** (critic) | **0.541** | **264.5** |
> | CAPS+PAVE | 0.543 | 264.0 |
>
> CAPS alone improves smoothness substantially over Base (sm 1.809→0.702). PAVE further improves upon this (sm 0.541) with higher return (264.5 vs 249.0). When combined, CAPS+PAVE (sm 0.543, re 264) performs comparably to PAVE alone, suggesting that once the critic geometry is stabilized, the actor naturally tracks the smooth implied policy without additional constraints. We will include this analysis in the revised appendix.
>
> **S3. Limitations**
>
> We are grateful to the reviewer for identifying these omissions. The revised manuscript will include an explicit limitations section:
>
> 1. **Training overhead**: PAVE adds ~1.0–1.4× wall-clock convergence time for TD3. This overhead is confined to training; inference speed is unaffected.
> 2. **Theoretical scope**: The proxy losses (Sec. 5) incentivize the desired geometric properties rather than guaranteeing them. The assumptions in Sec. 4.1 serve as local regularity conditions.

---

> > ### Author Rebuttal · Reviewer_cvdZ · 2026-04-02
> >
> > Thank you for the clarification. I will maintain my positive rating.

---

> > > ### Author Response · Authors · 2026-04-03
> > >
> > > Dear Reviewer cvdZ,
> > >
> > > Thank you for maintaining your positive rating and for acknowledging that the concerns have been fully resolved.
> > >
> > > Your two questions — on the distinct roles of MPR and VFC, and on the sufficiency of critic-side regularization — led us to conduct analyses that we believe strengthen the paper. The factorial ablation revealed that $\mathcal{L}_\text{Curv}$ is a prerequisite for the other components to be effective, and the CAPS+PAVE comparison provided direct evidence that critic geometry regularization alone is sufficient for achieving smooth policies. Both findings will be included in the revised manuscript.
> > >
> > > We also appreciate your comment on the missing limitations discussion, which prompted us to add an explicit section addressing training overhead and theoretical scope.
> > >
> > > Thank you for your careful reading and for recognizing PAVE as a fresh perspective. Your feedback has been valuable in refining our presentation.
> > >
> > > Best regards,
> > > Authors of Submission 15092

---

### Official Review · Reviewer_5bbp · 2026-03-04

**Soundness:** 2
**Presentation:** 3
**Significance:** 2
**Originality:** 3
**Overall Recommendation:** 5
**Confidence:** 3

**Summary:**

This paper identifies the critic’s geometry as the primary source of action instability and proposes the Policy Aware Value-field Equalization (PAVE), a critic- centric regularization framework that treats the critic as a scalar field and stabilizes its induced action-gradient field.

**Compliance With Llm Reviewing Policy:**

Affirmed.

**Final Justification:**

I will keep my score.

**Key Questions For Authors:**

- How and where does the Lipschitz bound derived be used in the algorithm? Specifically, do we need to estimate M or how we can get M? I kind of get the idea of M comes from the MPR but am not sure about it.
- How to choose the hyper parameters in equation 16?

**Limitations:**

Yes.

**Strengths And Weaknesses:**

## Strengths
- This paper proves that the non-smoothness comes from the Hessian structure and the curvatures
- Numerical experiments show that PAVE improves the smoothness and robustness.


## Weaknesses
- The proof relies on the assumption on curvatures, I'm wondering if in real life situations if that condition can be made.
- The paper claims PAVE improve the smoothness however the experiments may not directly illustrate that (Sec 6.4 is on robustness).

---

> ### Author Rebuttal · Authors · 2026-03-30
>
> We thank the reviewer for the thoughtful questions. We address each below.
>
> **S1. Reframing Sec. 6.4 and Smoothness Metric**
>
> We acknowledge that the term "robustness" in our manuscript may have caused confusion. Our intent was to show that PAVE's smooth policies achieve competitive performance even under noisy observations, not to claim robustness as a separate contribution. To address this confusion, we will correct the following in the revised manuscript:
>
> - **Abstract**: "smoothness and robustness comparable to..." → "smoothness comparable to..."
> - **Sec 6.4 title**: "Robustness to Observation Noise" → "Evaluation under Noisy Observations"
> - **Sec 6.4 body**: "We evaluated robustness against..." → "We evaluated performance under..."
> - **Table 3 caption**: "Robustness under varying..." → "Performance under varying..."
>
> Regarding the reviewer's concern that experiments may not directly illustrate smoothness: Tables 1–2 report the smoothness score (sm), a spectral metric derived via FFT analysis following Mysore et al. (2021) and Christmann et al. (2024). PAVE achieves the best or competitive sm across both TD3 and SAC settings. This is the central empirical evidence for smoothness.
>
> **S2. The Role of M in the Algorithm**
>
> We appreciate the reviewer's question, which highlights that the connection between the theoretical bound and the algorithm deserves clearer exposition. We recognize that our manuscript did not make this link sufficiently explicit, and we are grateful for the opportunity to clarify.
>
> The bound L ≤ M/μ (Prop. 4.2) is not computed at runtime. Directly computing M requires constructing the full mixed Hessian at every training step, which is prohibitive. Instead, PAVE's losses indirectly suppress M:
>
> - **MPR** (Eq. 11): perturbs states with random noise ε and penalizes Q-gradient changes. By Taylor expansion, this approximates ‖∇²_sa Q‖², reducing M.
> - **VFC** (Eq. 13): applies the same principle along actual transitions Δs, targeting M along dynamics-relevant directions.
>
> The reviewer's intuition is correct: **M comes from MPR** (and VFC). To verify this, we measured M_sup on trained models by computing the full mixed Hessian via autograd and extracting its largest singular value at each timestep:
>
> | | Lunar | Pend. | Reach. | Ant | Hop. | Walk. |
> |---|---|---|---|---|---|---|
> | Base | 990 | 447 | **16** | 8067 | 8831 | 1923 |
> | ASAP | 1227 | 393 | 17 | 7964 | 27627 | 1772 |
> | **PAVE** | **151** | **210** | 26 | **5962** | **934** | **643** |
>
> PAVE achieves the lowest M_sup in 5 out of 6 environments (e.g., Lunar 990→151, 6.6× reduction). Policy-side methods (ASAP) do not reduce M and can even increase it (Hopper: 8831→27627), because they regularize the actor without addressing the critic geometry that generates M.
>
> We will revise Sec. 5 to make this proxy→M connection more explicit.
>
> **S3. Hyperparameter Selection**
>
> The three loss weights λ₁ (MPR), λ₂ (VFC), λ₃ (Curv) in Eq. 16 are selected via coordinate search: starting from λ₁=0.1, λ₂=0.1, λ₃=0.01, each weight is swept one-at-a-time. The tuning order is λ₃ first (most sensitive), then λ₂, then λ₁.
>
> Sensitivity sweep on Lunar and Walker (sm↓ / re↑), conducted on different hardware from the submitted manuscript:
>
> | Scale | λ₃ (Curv) sm | λ₂ (VFC) sm | λ₁ (MPR) sm |
> |---|---|---|---|
> | ×0.01 | 1.35 / 1.78 | 1.13 / 1.70 | 0.63 / 1.61 |
> | ×1 | **0.54 / 1.52** | **0.54 / 1.52** | **0.54 / 1.52** |
> | ×100 | 0.32 / 1.31 | 0.47 / 1.27 | 0.60 / 1.80 |
>
> | Scale | λ₃ (Curv) re | λ₂ (VFC) re | λ₁ (MPR) re |
> |---|---|---|---|
> | ×0.01 | 250 / 4753 | 156 / 5142 | 262 / 4728 |
> | ×1 | **255 / 5383** | **255 / 5383** | **255 / 5383** |
> | ×100 | 261 / 4253 | 264 / 4480 | 257 / 4598 |
>
> λ₃ is most sensitive (sm range 1.03), λ₁ most stable (0.09). While increasing λ₃ to ×100 further reduces sm, it degrades return (Walker: 5383→4253).
>
> **S4. Assumption Verification**
>
> We computed the full eigenvalue decomposition of ∇²_aa Q at intermediate checkpoints via exact autograd. Neg Def Rate: fraction of (s,a) where all eigenvalues < 0.
>
> LunarLander:
>
> | Step | | Neg Def Rate↑ |
> |---|---|---|
> | 100K | Base / PAVE | 0.277 / **0.497** |
> | 300K | Base / PAVE | 0.283 / **0.689** |
> | 500K | Base / PAVE | 0.332 / **0.808** |
>
> Walker:
>
> | Step | | Neg Def Rate↑ |
> |---|---|---|
> | 200K | Base / PAVE | 0.071 / **0.548** |
> | 600K | Base / PAVE | 0.121 / **0.600** |
> | 1M | Base / PAVE | 0.144 / **0.638** |
>
> Base's concavity satisfaction remains low even at convergence (65–86% of states ill-conditioned). PAVE's L_Curv improves satisfaction to 64–84% across environments. Despite not reaching 100%, PAVE achieves the best smoothness in all 6 TD3 environments (Tables 1–2). Full results will be included in the revised Appendix.

---

> > ### Author Rebuttal · Reviewer_5bbp · 2026-04-01
> >
> > Thank you for clarifying my concerns. I have raised my score.

---

> > > ### Author Response · Authors · 2026-04-03
> > >
> > > Dear Reviewer 5bbp,
> > >
> > > Thank you for your acknowledgement and for raising your score. We are glad that the clarifications on the role of $M$, the hyperparameter selection procedure, and the assumption verification addressed your concerns.
> > >
> > > Your question on how the Lipschitz bound connects to the algorithm was particularly valuable — it prompted us to make the proxy→$M$ link more explicit in the revised manuscript. We will ensure that the revised Sec. 5 and appendix clearly reflect this improvement.
> > >
> > > We also appreciate your observation regarding Sec. 6.4's framing. Renaming it to "Evaluation under Noisy Observations" and removing the robustness claim from the abstract will prevent the confusion you identified, making the paper's contributions more precisely stated.
> > >
> > > Thank you again for your constructive feedback and for recognizing the theoretical contribution of our work. Your questions directly led to improvements in both exposition and empirical analysis that will be reflected in the camera-ready version.
> > >
> > > Best regards,
> > > Authors of Submission 15092

---

### Official Review · Reviewer_qWNV · 2026-03-12

**Soundness:** 4
**Presentation:** 3
**Significance:** 3
**Originality:** 3
**Overall Recommendation:** 5
**Confidence:** 3

**Summary:**

This paper studies the causes and solutions for "jerky" control signals in actor-critic methods. The authors first mentioned that actor-critic policies often exhibit oscillatory behavior, and argue that this stems from the differential geometry of the critic's value landscape. Building on this theoretical insight, the paper proposes PAVE, a regularization framework that stabilizes policy actions by constraining the geometric properties of the critic. Experimental results on high-dimensional MuJoCo environments show that PAVE achieves smooth control while maintaining competitive task return.

**Compliance With Llm Reviewing Policy:**

Affirmed.

**Final Justification:**

The authors' rebuttal addressed my main concerns and clarified that certain assumptions, such as S3 in rebuttal, may not always hold. This represents a minor theoretical weakness that worth future exploration. In general,  the paper remains strong overall. I lean toward acceptance.

**Key Questions For Authors:**

1. It is beneficial to statistically measure gradient direction flips (e.g., cosine similarity of consecutive Q-gradients) in a naive actor-critic versus PAVE, to provide more direct evidence for the geometric instability claim.
2. It is beneficial to report wall-clock convergence time with and without PAVE across the benchmark environments.
3. How often do the theoretical assumptions  hold during actual training?

**Limitations:**

The authors did not explicitly discuss limitations. The computational cost might be one of the limitations.

**Strengths And Weaknesses:**

**Strengths:**
1. The paper identifies that jerky control is related to the geometric structure of the critic's value function that guides the actor, rather than the actor itself. This is an interesting and compelling finding.
2. The proposed method, PAVE is directly motivated by the theoretical insight — each regularization component maps to a specific term in the Lipschitz bound. The framework is reasonable and well-justified.
3. Overall, the paper is well-structured and clear.

**Weaknesses:**

1. The paper claims that the Q-gradient field is "geometrically unstable" and that gradient directions flip drastically with small state perturbations. It would be beneficial to statistically measure gradient direction flips in a naive actor-critic versus with the proposed regularization, providing more direct evidence for the core claim.

2. The proposed regulaorzation PAVE adds substantial computational cost and training time. The paper should report convergence time with and without the proposed method.

3. Several key assumptions , including the negative definiteness of action Hessian, bounded mixed partial derivatives, and concavity of action Hessian, may not hold in practice with neural Q-functions. The paper should discuss when these assumptions are likely to be hold.

---

> ### Author Rebuttal · Authors · 2026-03-30
>
> We thank the reviewer for recognizing PAVE's theoretical insight and well-structured framework. We address each suggestion below.
>
> **S1. Statistical Evidence for Q-Gradient Instability**
>
> We appreciate the reviewer's suggestion of measuring cosine similarity between consecutive Q-gradients, which provides direct evidence for our geometric instability claim. On trained models, we computed ∇_a Q via exact autograd at each timestep and measured the cosine similarity between consecutive pairs (∇_a Q(s_t, a_t), ∇_a Q(s_{t+1}, a_{t+1})). A negative cosine indicates a gradient direction flip, meaning the critic provides contradictory update signals to the actor at adjacent states.
>
> Mean cosine similarity (↑) and flip rate (cos < 0, ↓):
>
> | Env | Base cos↑ | PAVE cos↑ | Base flip↓ | PAVE flip↓ |
> |---|---|---|---|---|
> | Lunar | 0.593 | **0.901** | 0.184 | **0.033** |
> | Pend. | 0.491 | **0.954** | 0.254 | **0.023** |
> | Reach. | 0.950 | **0.968** | 0.016 | **0.010** |
> | Ant | 0.153 | **0.183** | 0.355 | **0.333** |
> | Hop. | 0.566 | **0.755** | 0.165 | **0.073** |
> | Walk. | 0.635 | **0.835** | 0.090 | **0.014** |
>
> PAVE reduces flip rates by 2 to 11 times in 4 out of 6 environments (Pendulum 25.4%→2.3%, Walker 9.0%→1.4%, Lunar 18.4%→3.3%, Hopper 16.5%→7.3%). Reacher shows moderate improvement (1.6%→1.0%) from an already low baseline flip rate. We will include this analysis in the appendix.
>
> **S2. Wall-Clock Convergence Time**
>
> We agree with this suggestion. From the same training runs in Tables 1–2, we measured wall-clock convergence time. Convergence is the first time the smoothed learning curve reaches the method's own final performance (last 10% mean). TD3 convergence time (minutes):
>
> | | Lunar | Pend. | Reach. | Ant | Hop. | Walk. |
> |---|---|---|---|---|---|---|
> | Base | 206 | 26 | 62 | 392 | 202 | 400 |
> | PAVE | 203 | 36 | 79 | 453 | 212 | 514 |
> | ratio | 0.98× | 1.38× | 1.27× | 1.15× | 1.05× | 1.29× |
>
> While PAVE's per-step throughput is lower (Table 5, ~3.9×), the wall-clock convergence ratios (1.0–1.4×) are substantially smaller because PAVE converges in fewer gradient steps in 4 out of 6 environments. Inference speed is identical. We will replace the Complexity Analysis in the revised manuscript.
>
> **S3. Assumption Satisfaction During Training**
>
> We thank the reviewer for pointing this out. We measured two quantities from Prop. 4.2 at intermediate checkpoints via exact autograd:
>
> - M_sup = sup ‖∇²_sa Q‖₂ (spectral norm): the numerator of the Lipschitz bound L ≤ M/μ, computed from the full d_a × d_s mixed Hessian.
> - Neg Def Rate: fraction of (s,a) points where all eigenvalues of ∇²_aa Q are strictly negative, directly measuring Assumption 4.1 (strict concavity).
>
> **During-training progression:**
>
> LunarLander:
>
> | Step | | M_sup↓ | Neg Def Rate↑ |
> |---|---|---|---|
> | 100K | Base / PAVE | 273 / **101** | 0.277 / **0.497** |
> | 300K | Base / PAVE | 616 / **129** | 0.283 / **0.689** |
> | 500K | Base / PAVE | 1105 / **188** | 0.332 / **0.808** |
>
> Walker:
>
> | Step | | M_sup↓ | Neg Def Rate↑ |
> |---|---|---|---|
> | 200K | Base / PAVE | 636 / **167** | 0.071 / **0.548** |
> | 600K | Base / PAVE | 1212 / **417** | 0.121 / **0.600** |
> | 1M | Base / PAVE | 1320 / **659** | 0.144 / **0.638** |
>
> Base's M_sup grows over training (Lunar: 273→1105, 4.0×) while PAVE remains stable (101→188, 1.9×). This reflects unconstrained critic training: as the Q-network fits complex return landscapes, ∇²_sa Q becomes more volatile. PAVE's MPR and VFC losses counteract this growth.
>
> Base's Neg Def Rate is low but increases over training (Lunar: 0.28→0.33, Walker: 0.07→0.14), suggesting that concavity around optimal actions is a natural property that emerges as Q-learning improves the value estimate. However, this natural progression is slow and insufficient — even at convergence, 65–86% of states remain ill-conditioned. PAVE's L_Curv accelerates this tendency, improving concavity to 0.81 (Lunar) and 0.64 (Walker). This theoretical scope will be explicitly stated in revised Sec. 4.1.
>
> **S4. Limitations**
>
> The revised manuscript will include an explicit limitations section:
>
> 1. **Training overhead**: PAVE adds ~1.0–1.4× wall-clock convergence time for TD3. This overhead is confined to training; inference speed is unaffected.
> 2. **Theoretical scope**: The proxy losses (Sec. 5) incentivize the desired geometric properties rather than guaranteeing them. The assumptions in Sec. 4.1 serve as local regularity conditions.
>
> We are grateful to the reviewer for identifying these omissions.

---

> > ### Author Rebuttal · Reviewer_qWNV · 2026-04-03
> >
> > The authors' rebuttal addressed my main concerns and clarified that certain assumptions may not always hold (e.g.,  S3). I recommend explicitly mentioning this as a limitation in the final version of the paper.

---

> > > ### Author Response · Authors · 2026-04-03
> > >
> > > Dear Reviewer qWNV,
> > >
> > > Thank you for your acknowledgement and for maintaining your positive score. We are glad that the cosine similarity analysis, wall-clock convergence time, and during-training assumption measurements addressed your concerns.
> > >
> > > Your suggestion to statistically measure Q-gradient direction flips provided direct empirical evidence for our core geometric instability claim, which we believe adds significant value to the paper. We also appreciate your recommendation to report convergence time — it led us to adopt wall-clock convergence as a more faithful complexity measure than FPS alone.
> > >
> > > As you recommended, we will explicitly state in the revised manuscript that the strict concavity assumption may not universally hold, and that $L \leq M/\mu$ serves as regularization motivation rather than a guarantee. This is reflected in the Limitations section and in two new Remarks added to Sec. 4.
> > >
> > > We appreciate your thoughtful evaluation, which has strengthened both the empirical rigor and the presentation of our work.
> > >
> > > Best regards, Authors of Submission 15092

---

### Official Review · Reviewer_vzUU · 2026-03-13

**Soundness:** 3
**Presentation:** 3
**Significance:** 3
**Originality:** 2
**Overall Recommendation:** 4
**Confidence:** 4

**Summary:**

The authors analyze the problem of policy smoothness in actor-critic frameworks. They argue that the primary source for an instable and brittle actor is the geometry of the critic and that applying policy smoothing as done in prior work is treating the symptoms rather than the cause of the issue. A theoretical examination using implicit differentiation of $Q$ derives a connection between policy sensitivity and second-order critic quantities. Motivated by this, the authors propose three additional regularization terms (PAVE) that aim to stabilize mixed partial derivatives and preserve local curvature of $Q$ with respect to $a$, with the goal of obtaining a smoother $Q$-gradient field and therefore a smoother policy. The paper presents both theory and empirical results on default continuous-control RL benchmarks.

**Compliance With Llm Reviewing Policy:**

Affirmed.

**Final Justification:**

I appreciate the solid effort put into this rebuttal. The authors have addressed my primary concerns by adding critic-centric baselines on 2 tasks, validating the proxy objectives against autograd-based Hessian quantities and clarifying the limits of their theoretical assumptions. It's good that the authors are willing to tone down the guarantee-style claims to more appropriate proxy/incentive language and to align the architectural choices with the SiLU reruns.

While I still feel some of the new evidence is a bit narrow in scope and a few of these fixes assume the authors will properly integrate them into the camera-ready version, my main issues are largely resolved. This leads to an increase of the general rating to weak accept, and the soundness and significance by one point.

**Key Questions For Authors:**

1.  Why is there no direct comparison to critic-centric baselines (architectural smoothness or gradient-penalty approaches on the critic) given that the main claim of the paper is that one should regularize the critic rather than smooth the policy? A stronger comparison here would affect my evaluation of the empirical contribution.

2.  How restrictive do the authors believe the strict local maximum / interior-point and strict local concavity assumptions in sec. 4.1 and prop. 4.2 are in the practical continuous-control setting considered in the experiments? how should one interpret the theoretical results when optimal actions lie on the boundary or are not unique? A discussion would help me understand how much of the theory carries over to the actual RL setting.

3.  Can the authors provide any empirical validation of the approximation quality used in sec. 5.1 / 5.2 for example by comparing the finite-difference proxy to actual Hessian-related quantities on some examples? This would make the bridge from theory to the implemented loss more convincing and would positively affect my view of the paper.

4.  The paper often phrases the proposed objectives as if they directly bound $M$ or enforce curvature properties while the derivations rely on local approximations and proxy losses. Are the authors willing to tone down these claims and distinguish more clearly between what is proven, approximated, and what is only incentivized?

5.  The proposed method uses SiLU activations while the baselines apparently do not and using SiLU for TD3 seems to hurt performance. Did the authors investigate this effect and can they report more comparison results with aligned architectural choices?

**Limitations:**

Yes

**Strengths And Weaknesses:**

The paper is well written and the theoretical sections are insightful. The link between policy smoothness and specific second-order properties of the critic is interesting. I think the general motivation is good, rather than smoothing the policy directly, the paper asks whether one should instead regularize the critic geometry that induces brittle actor updates which is a relevant question.

However, I am not fully convinced that the strength of the theoretical part matches what is actually implemented and encountered in real setups. The analysis in sec. 4.1 relies on a strict local maximum / interior-point assumption for $a^{\star}(s)$ and similarly prop. 4.2 assumes strict local concavity of the critic in action $a$. While this is useful for the implicit derivative argument, it seems to me quite restrictive in practical continuous-control problems, where optimal actions can lie on the boundary and also not be unique.
The bound $L \leq M / \mu$ is fine under the here stated assumptions, but the assumption $\lambda_{\max}(\nabla_{aa}^2 Q_\theta(s,a^*(s))) \leq -\mu < 0$ seems quite strong for a neural network critic. The paper should comment more clearly on how restrictive these assumptions are in the actual practical RL setup.

the nice theoretical analysis gets approximated in sec. 5, which is fine, but then the wording should reflect this more carefully in my opinion. In sec. 5.1, Eq. (12) is obtained after linearizing the actual objective and dropping higher-order terms. The implemented loss in Eq. (11) is a finite-difference proxy and not the mixed Hessian itself. Therefore the claim that minimizing Eq. (11) bounds the spectral norm $M$ seems too strong as written. It encourages a smaller upper bound on $M$ but this depends on how accurate the approximation is for the chosen perturbation scale $\epsilon$.

a similar issue appears in sec. 5.2. where the intuition of temporal consistency is clear but the formal interpretation seems a bit too strong to me. The loss in Eq. (13) penalizes differences between $\nabla_a Q_\theta(s_t,a_t)$ and $\nabla_a Q_\theta(s_{t+1},a_t)$ along sampled transitions. This might encourage a smoother gradient field over time but I am not convinced by the claim that this is analogous to minimizing a Fisher divergence in a rigorous sense. The connection to the mixed Hessian in Eq. (14) relies on a first-order approximation $s_{t+1} \approx s_t + \Delta s_t$, so this is a local approximation. Therefore statements such as bounding $M$ along the system manifold or ensuring kinematic realizability seem stronger than what is actually shown here.

in sec. 5.3, Curv is described as penalizing the trace of the action Hessian but the proposed loss is actually a hinge loss on sampled directional curvatures $v^\top \nabla_{aa}^2 Q_\theta(s,a) v$. From this, I do not think one can directly conclude that the method enforces $\lambda_{\max}(\nabla_{aa}^2 Q_\theta) \leq -\delta$. It again incentivizes but does not guarantee it.

On the empirical side, the comparison does not fully isolate where the gains come from. The comparison with prior work focuses on methods that use policy smoothing (CAPS, GRAD, ASAP) while the paper argues that this is not what one should do. I therefore think the paper should compare more directly against critic-centric baselines.
The authors already mention related techniques in sec. 2.2 on critic geometry and gradient regularization. Even if these methods do not explicitly connect critic differential geometry to policy smoothness, they are much closer to what is done here. A simple baseline should be included to compare the authors more sophisticated approach to methods that just try to obtain a smoother $Q$ via architectural smoothness, gradient penalties, etc.

I was missing some attempt to validate the approximation quality for example in sec. 5.1 instead of only relying on the finite-difference. The authors could have tried to actually compute Hessian-related quantities (using a differentiable simulators for RL) for at least some examples and compare them to the approximation. This would make the bridge from theory to method more convincing.

there are some comparison issues in the implementation. The authors use SiLU activation functions in their framework but not for the other baselines. I think a fair comparison that aims to remove doubts should also run the baselines with this activation function, to reduce the difference as much as possible. Using SiLU for the TD3 baseline made the performance worse, going from re 244 to 225 and sm 1.4 to 1.8. I think the paper should discuss this more

some additional minor comments: authors show ablation by adding one term after another, but unfortunately we do not see the combination of just MPR + CURV, which might be interesting given that VFC leads to a drop for walker. For all curvature plots are the plots all on the same scale? Also, why is there a comma between $s$ and $a$ in the axis label, while usually one just writes $\nabla_{sa}^2 Q$. Please pull the information that experiments were run with 5 seeds from the appendix into the main table. It would be good to say std over 5 seeds directly in the header. In gymnasium mujoco env "walker" does not exist it should be called explicitly walker2D. why did the authors stop there and not try humanoid? There is no discussion of action chunking, although this is one of the more direct ways to obtain temporally more consistent actions, so at least some discussion in related work could be useful.

---

> ### Author Rebuttal · Authors · 2026-03-30
>
> We appreciate the reviewer's thorough and constructive feedback.
>
> **S1. Comparison with Critic-Centric Baselines**
>
> We agree with the reviewer's valid comment. We implemented **SN-Critic** (spectral normalization; Gogianu et al., 2021) and **GP-Critic** (gradient penalty λ‖∇_a Q‖²) and compared under TD3 with SiLU on Lunar and Walker, measuring sm/re alongside M_sup (exact autograd; Prop. 4.2 numerator) and negative definiteness rate (Prop. 4.2 denominator condition):
>
> | Method | sm↓ | re↑ | M_sup↓ | Neg Def↑ |
> |---|---|---|---|---|
> | Base | 1.81 / 1.99 | 228 / 4834 | 990 / 1923 | 0.35 / 0.14 |
> | SN-Critic | 0.00 / 0.00 | −970 / 14 | 0.1 / − | 0.40 / 0.18 |
> | GP-Critic | 2.18 / 1.66 | 202 / 5011 | 882 / 1090 | 0.37 / 0.15 |
> | **PAVE** | **0.54** / **1.27** | **264** / **5563** | **151** / **643** | **0.84** / **0.64** |
>
> SN-Critic collapses entirely (re: −970 / 14), producing near-constant actions (sm 0.00). Spectral normalization suppresses both M and μ indiscriminately, destroying the critic's learning signal. GP-Critic reduces M_sup moderately (Lunar: 990→882, Walker: 1923→1090) but does not improve μ (Neg Def: 0.37 / 0.15, no different from Base). Its sm worsens on Lunar (1.81→2.18) and improves only slightly on Walker (1.99→1.66). PAVE is the only method that reduces M while simultaneously improving μ (Neg Def: 0.84 / 0.64), achieving the best smoothness among successfully trained methods (0.54 / 1.27) and the highest return (264 / 5563).
>
> **S2. Scope of Theoretical Assumptions**
>
> We acknowledge that the interior-point and strict concavity assumptions in Lemma 4.1 do not hold universally. When optimal actions lie on the boundary, the implicit function theorem does not apply; when optimal actions are not unique, the policy Jacobian in Eq. 2 is not well-defined. The bound L ≤ M/μ in Prop. 4.2 is valid only where these conditions are met. In revised Sec. 4, we will add:
>
> - **Lemma 4.1**: a **Remark** that the result does not apply at boundary optima or non-unique a*(s).
> - **Prop. 4.2**: a **Remark** that λ_max ≤ −μ < 0 may not hold everywhere for neural critics, and that L ≤ M/μ serves as regularization motivation rather than a universal guarantee.
>
> **S3. Proxy Validation**
>
> We validated MPR (Eq. 11 ≈ Eq. 12) and VFC (Eq. 13 ≈ Eq. 14) by computing both the proxy loss and the Hessian-related quantity via exact autograd on trained trajectories:
>
> | Env | MPR Ratio | VFC Ratio | M_sup (Base → PAVE) |
> |---|---|---|---|
> | Pend. | 1.14 | 0.91 | 447 → 210 |
> | Hop. | 1.04 | 0.30 | 8831 → 934 |
> | Walker | 0.91 | 0.31 | 1923 → 643 |
>
> MPR ratios are reasonable (full range:0.67–1.14). VFC ratios degrade when ‖Δs‖ is large, as the proxies are designed to reduce M, not to replicate the Hessian exactly. Indeed, PAVE reduced M_sup by 2.1–9.5×, confirming that the proxies fulfill their intended role. We will include this analysis in the revised appendix.
>
> **S4. Refining Theoretical Claims**
>
> In revised Sec. 5, we will add: "The following losses are finite-difference proxies that incentivize the desired geometric properties rather than guaranteeing them." Specific revisions:
>
> - **Sec 5.1**: "bounds the spectral norm M" → "encourages a smaller M"
> - **Sec 5.2**: "bound the mixed-partial term M along the manifold" / "ensuring kinematic realizability" → "encourage smaller M along dynamics-relevant directions" / removed
> - **Sec 5.3**: "stochastically enforces the spectral constraint λ_max ≤ −δ" → "incentivizes via Hutchinson's trace estimator, which controls the trace rather than individual eigenvalues"
>
> **S5. Comparison under Aligned Activation**
>
> PAVE requires SiLU because L_Curv needs C²-continuous activations for Hutchinson's estimator. We agree with the concern and re-ran baselines under SiLU. TD3 smoothness (sm↓):
>
> | | Lunar | Pend. | Reach. | Ant | Hop. | Walk. |
> |---|---|---|---|---|---|---|
> | Base | 1.81 | 1.59 | 0.053 | 2.04 | 2.72 | 1.99 |
> | ASAP | 2.06 | 1.97 | 0.051 | 2.07 | 1.99 | 1.62 |
> | **PAVE** | **0.54** | **0.35** | **0.039** | **1.77** | **0.95** | **1.27** |
>
> PAVE maintained the best smoothness in all 6 TD3 environments with competitive return, confirming that activation choice did not significantly affect relative performance. Tables 1–2 will use SiLU-unified results.
>
> **S6. Ablation, Humanoid, and Minor Revisions**
>
> Requested ablation on Walker (SiLU, TD3), conducted on different hardware from the submitted manuscript:
>
> | Config | Walker sm↓ | Walker re↑ |
> |---|---|---|
> | Base | 1.828 | 4589 |
> | MPR+VFC | 2.095 | 5010 |
> | MPR+Curv | 1.718 | 5177 |
> | **Full PAVE** | **1.483** | **5502** |
>
> L_Curv is decisive: without it, sm worsened beyond Base. We initially excluded Humanoid because Base TD3 fails to learn in this environment. Following the reviewer's suggestion, we extended to Humanoid and confirmed that PAVE enables TD3 to learn in this environment (re: 309 → 2894, sm: 0.06 → 0.76; low Base sm reflects near-zero learning). Action chunking and notation issues will be addressed in the revised manuscript.

---

> > ### Author Rebuttal · Reviewer_vzUU · 2026-04-03
> >
> > I appreciate the solid effort put into this rebuttal. The authors have addressed my primary concerns by adding critic-centric baselines on 2 tasks, validating the proxy objectives against autograd-based Hessian quantities and clarifying the limits of their theoretical assumptions. It's good that the authors are willing to tone down the guarantee-style claims to more appropriate proxy/incentive language and to align the architectural choices with the SiLU reruns.
> >
> > While I still feel some of the new evidence is a bit narrow in scope and a few of these fixes assume the authors will properly integrate them into the camera-ready version, my main issues are largely resolved.
> >
> > minor note: the rebuttal mentions the MPR ratios have a full range of 0.67 - 1.14 but the lowest MPR ratio in the provided table is 0.91 (for Walker). It looks like 0.67 might be a typo or it refers to an environment not included in the table.

---

> > > ### Author Response · Authors · 2026-04-03
> > >
> > > Dear Reviewer vzUU,
> > >
> > > We sincerely appreciate the nature of your review throughout this process. Your feedback on critic-centric baselines, proxy validation, and theoretical overclaiming has substantially improved our work. In particular, the suggestion to compare against SN-Critic and GP-Critic led to an important addition that directly validates our theoretical bound — that reducing $M$ alone (without preserving $\mu$) is insufficient, which is precisely what PAVE's $\mathcal{L}_\text{Curv}$ addresses.
> > >
> > > Regarding your minor note: thank you for catching this. The 0.67 refers to LunarLander, which was not included in the abbreviated table.
> > >
> > > We will include the complete results in the revised appendix.
> > >
> > > We are grateful for your thorough evaluation, which has significantly improved both the presentation and empirical rigor of our work. All proposed revisions — toned-down theoretical claims, SiLU-unified Tables 1–2, proxy validation appendix, and explicit limitation section — will be reflected in the camera-ready version.
> > >
> > > Best regards,
> > > Authors of Submission 15092

---

### Decision · Program_Chairs · 2026-04-30

**Decision:**

Accept (spotlight)

**Comment:**

This paper studies policy smoothness in continuous actor–critic methods and argues that non-smooth (oscillatory) policies stem from the differential geometry of the critic rather than the actor itself. Building on this insight, the authors derive a theoretical relationship linking policy sensitivity to second-order properties of the Q-function, and propose PAVE, a critic-centric regularization framework that stabilizes the Q-gradient field. Empirical results on standard continuous-control benchmarks show improved smoothness with competitive returns.

All reviewers find the problem important and the perspective novel. In particular, the idea of addressing policy smoothness through critic geometry, rather than direct policy regularization, is considered insightful and well-motivated. The paper is generally regarded as clearly written, with a coherent link between theory and method, and supported by reasonably strong empirical performance. Several reviewers also highlight that the proposed components of PAVE are well aligned with the theoretical formulation.

The main concerns focus on (i) the restrictiveness of theoretical assumptions (e.g., local concavity and interior optimality), (ii) the gap between theoretical claims and the proxy objectives used in practice, and (iii) empirical evaluation completeness, including comparisons with critic-centric baselines and analysis of computational overhead. These issues were taken seriously by the authors in the rebuttal. They provided additional experiments (including critic-centric baselines and proxy validation against autograd-computed quantities), clarified the scope of assumptions, and toned down several overstatements to better reflect that the proposed losses act as incentives rather than guarantees. Reviewers acknowledged that these revisions largely address their concerns, although some evidence remains somewhat limited in scope.

Overall, the paper presents a conceptually and practically solid contribution to reinforcement learning, with a perspective that could inspire further work on critic-aware regularization.